# Dual role for Jumu in the control of hematopoietic progenitors in the *Drosophila* lymph gland

**Yangguang Hao, Li Hua Jin\***

Department of Genetics, College of Life Sciences, Northeast Forestry University, Harbin, China

**Abstract** The *Drosophila* lymph gland is a hematopoietic organ in which the maintenance of hematopoietic progenitor cell fate relies on intrinsic factors and extensive interaction with cells within a microenvironment. The posterior signaling center (PSC) is required for maintaining the balance between progenitors and their differentiation into mature hemocytes. Moreover, some factors from the progenitors cell-autonomously control blood cell differentiation. Here, we show that Jumeau (Jumu), a member of the forkhead (Fkh) transcription factor family, controls hemocyte differentiation of lymph gland through multiple regulatory mechanisms. Jumu maintains the proper differentiation of prohemocytes by cell-autonomously regulating the expression of Col in medullary zone and by non-cell-autonomously preventing the generation of expanded PSC cells. Jumu can also cell-autonomously control the proliferation of PSC cells through positive regulation of dMyc expression. We also show that a deficiency of *jumu* throughout the lymph gland can induce the differentiation of lamellocytes via activating Toll signaling.

## Introduction

The development and specification of blood cells in humans and *Drosophila* are controlled by conserved transcriptional regulators and signaling pathways. *Drosophila* has become a suitable model for investigating the genetic and molecular mechanisms of hematopoiesis and associated leukemias (*Crozatier and Vincent, 2011*). Similarly to the process in vertebrates, hematopoiesis in *Drosophila* occurs in two waves during development. The first population of hemocytes is derived from the embryonic head mesoderm and provides two types of circulating blood cells, specifically plasmatocytes and crystal cells (*Holz et al., 2003*). The second wave of hematopoiesis occurs during the larval stage in a specialized hematopoietic organ known as the lymph gland (*Lanot et al., 2001*). At the onset of metamorphosis, the lymph gland breaks down, releasing plasmatocytes and crystal cells into the circulating hemolymph (*Crozatier and Vincent, 2011*). In response to an immune challenge, the lymph gland can also give rise to a third cell type, the lamellocytes, which are produced in response to wasp infestation (*Lanot et al., 2001*; *Crozatier and Meister, 2007*). The third-instar larval lymph gland contains a pair of primary lobes and a variable number of secondary lobes posterior to the primary lobes. The primary lobes are organized in three zones: the cortical zone (CZ) contains differentiated hemocytes, the medullary zone (MZ) is composed of prohemocytes, and the posterior signaling center (PSC) functions as a hematopoietic niche for maintaining a population of blood cell precursors (*Jung et al., 2005*). Similar to the hematopoietic stem cell (HSC) niche in the mammalian bone marrow, the PSC plays a key role in supporting *Drosophila* blood cell homeostasis (*Krzemień et al., 2007*; *Mandal et al., 2007*). PSC cells are specified by the expression of two transcription factors, the homeotic protein Antennapedia (Antp) and Collier (Col), which is the *Drosophila* early B-cell factor (EBF) ortholog (*Crozatier et al., 2004*; *Mandal et al., 2007*). In addition, PSC

**\*For correspondence:** lhjin2000@
hotmail.com

**Competing interests:** The authors declare that no competing interests exist.

cells also selectively express the secreted factor Hedgehog (Hh) and the Serrate (Ser) signaling molecules (*Lebestky et al., 2003*; *Krzemień et al., 2007*; *Mandal et al., 2007*). In the *col* mutant, all Antp-positive PSC cells are missing, and the population of MZ hematopoietic precursors is lost (*Mandal et al., 2007*). Conversely, increasing the PSC size via *Antp* overexpression can increase the size of the MZ at the expense of the CZ during late stages of the third instar (*Mandal et al., 2007*). Moreover, the decapentaplegic/bone morphogenetic protein (Dpp/BMP), Wingless (Wg) and insulin/TOR signaling pathways cell-autonomously control the size of the PSC by regulating the expression of dMyc (*Pennetier et al., 2012*; *Sinenko et al., 2009*; *Benmimoun et al., 2012*; *Tokusumi et al., 2015*). A recent study suggested that Slit/Robo signaling from the cardiac tube also controls PSC morphology (*Morin-Poulard et al., 2016*). Prohemocytes within the MZ express the Hh receptor Patched (Ptc), the Jak/Stat signaling pathway receptor Domeless (Dome), and DE-cadherin (Shg), which is a target of Wg function in prohemocytes (*Mandal et al., 2007*; *Sinenko et al., 2009*). Furthermore, a significant up-regulation of ROS levels occurs during the third larval instar, and this up-regulation primes hematopoietic progenitors for differentiation (*Owusu-Ansah and Banerjee, 2009*). It has been shown that the Hh signal from the PSC contributes to the maintenance of prohemocytes within the MZ (*Mandal et al., 2007*). In addition, Wingless and insulin/TOR signaling can autonomously maintain hemocyte progenitors in the lymph gland (*Sinenko et al., 2009*; *Benmimoun et al., 2012*).

Jumeau (Jumu) is a member of the winged-helix/forkhead (Fkh) transcription factor family in *Drosophila*. Previous studies have shown that Jumu is a suppressor of position-effect variegation and is involved in neurogenesis and in eye, wing and bristle development (*Strödicke et al., 2000*; *Cheah et al., 2000*). A recent study showed that Jumu and its homolog, Checkpoint suppressor homologue (CHES-1-like), regulate the division of cardiac progenitor cells through a Polo-dependent pathway (*Ahmad et al., 2012*). Our previous study demonstrated that the loss of *jumu* affects the number and phagocytosis of circulating hemocytes (*Jin et al., 2008*). Moreover, the overexpression of *jumu* induces melanotic nodules by activating Toll signaling (*Zhang et al., 2016*). In this paper, we show that Jumu is expressed in the entire lymph gland and plays crucial roles during *Drosophila* hematopoiesis. Jumu protein within the entire lymph gland suppresses the activation of Toll signaling by down-regulating Col expression and thereby prevents lamellocyte differentiation. Jumu protein within the MZ maintains the proper differentiation of prohemocytes by cell-autonomously preventing the local overexpression of Col in the MZ and by non-cell-autonomously inhibiting the ectopic and expanded Col$^+$ PSC cells. Moreover, our results also demonstrate that Jumu cell-autonomously controls PSC cell proliferation by regulating the expression of dMyc.

## Results

### Jumu is involved in the differentiation of hemocyte precursors

To further analyze the function of Jumu in larval hematopoietic homeostasis, we first detected the expression of the plasmatocyte marker P1, the crystal cell marker Hindsight (Hnt) and the lamellocyte marker L1 in the lymph glands of *jumu* mutants. Because all *jumu* homozygous null mutants died during embryogenesis, to further reduce the expression of Jumu, we utilized the double heterozygotes generated by crosses between different *jumu* heterozygote mutants. Only a few *jumu* double heterozygotes mutants survived until adulthood, and these were found to exhibit poor viability and female sterility. The *jumu* heterozygote mutant lymph gland showed obvious reductions in plasmatocyte and crystal cell differentiation (*Figure 1A–J*). Moreover, the number of crystal cells in the *jumu* double heterozygotes mutants was clearly less than that found in heterozygotes (*Figure 1I,J*). However, plasmatocyte differentiation in the *jumu* double heterozygotes mutants was not clearly reduced compared with that found in $w^{1118}$ (*Figure 1D,E*). In addition, only very few lamellocytes could be detected in less than 10% of the lymph glands (n = 30 lobes) of *jumu* heterozygotes, whereas more than 90% of the lymph glands in the *jumu* double heterozygotes mutants (n = 40 lobes) showed large numbers of lamellocytes in the primary and secondary lobes (*Figure 1K–N*). Moreover, a reduced primary lobe size and an enormous overgrowth of secondary lobes were features of *jumu* double heterozygotes lymph glands (*Figure 2E,L*). To further determine whether the change in hematopoietic homeostasis observed in the *jumu* double heterozygotes mutants depends on the expression levels of *jumu*, the transcription levels of *jumu* in heterozygotes and double

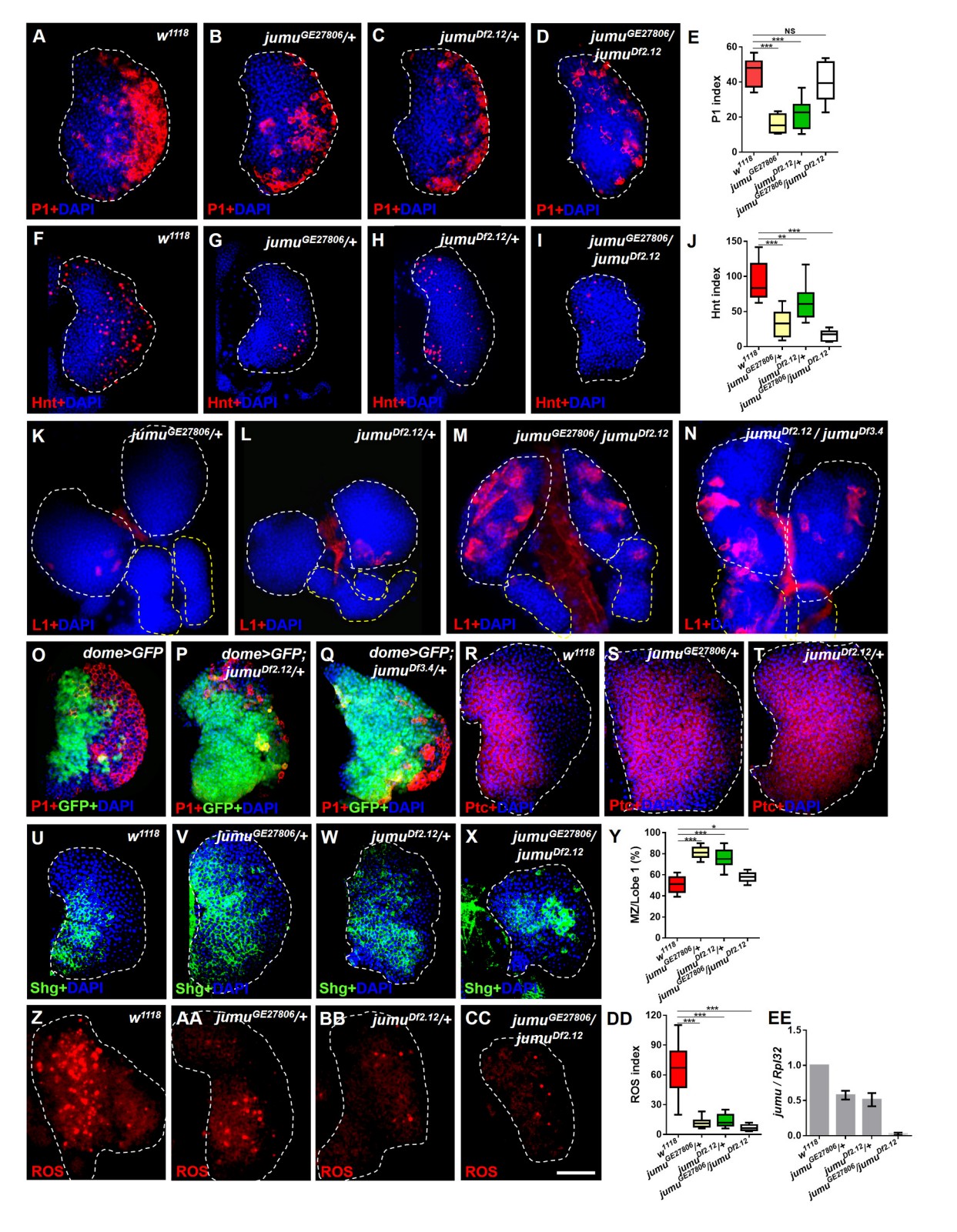

**Figure 1.** Loss of *jumu* affects prohemocyte differentiation in the lymph gland. (A–D) Immunostaining against the plasmatocyte marker P1 (red) in the lymph glands of third-instar larvae. The numbers of plasmatocytes are significantly reduced in *jumu* heterozygotes but not obviously reduced in double heterozygotes mutants. (F–I) Immunostaining against the crystal cell marker Hnt (red) in the lymph glands of third-instar larvae. The numbers of crystal cells are significantly reduced in the mutant flies. (E, J) Quantification of plasmatocyte and crystal cell indexes in primary lobes. (K–N) Immunostaining

*Figure 1 continued on next page*

*Figure 1 continued*

against the lamellocyte marker L1 (red) shows few lamellocytes in *jumu* heterozygotes (**K, L**) and substantial increases in the lamellocyte number in *jumu* double heterozygotes mutants (**M, N**). (**O–Q**) The lymph glands of *jumu*$^{Df2.12}$ and *jumu*$^{Df3.4}$ third-instar larvae display expansion of the medullary zone (MZ) as marked by dome>GFP (green) and a reduced plasmatocyte signal (red). (**R–X**) Immunostaining against Ptc (red) (**R–T**) and Shg (green) (**U–X**) shows MZ expansion in mutant third-instar larvae. (**Y**) Quantification of the proportions of the primary lobes occupied by prohemocytes (Shg$^+$ MZ area/ primary lobe area). (**Z–CC**) The ROS levels (red) are increased in the MZ of *w*$^{1118}$ third-instar larvae, but similar increases are rarely observed in mutant third-instar larvae. (**DD**) Quantification of the ROS$^+$ cell index in primary lobes. (**EE**) Real-time PCR analysis of the *jumu* level in the entire third-instar larvae. The data are presented as box-and-whisker representations in E, J, Y and DD and as column-bar-graph representations (in which the error bars represent ± S.E.M.) in EE. For all quantifications: NS, not significant; *p<0.1; **p<0.01; ***p<0.001 (Student's *t* test). Dashed white and yellow lines outline the edges of the primary and secondary lobes, respectively, in all of the figures. Scale bars: 50 μm.

heterozygotes third-instar larvae were quantified. The *jumu* transcription levels were reduced by approximately half and 40-fold in the heterozygotes and the double heterozygotes mutants, respectively (*Figure 1EE*). In summary, these results indicate that a change in *jumu* expression substantially disturbs blood cell homeostasis and the growth and development of the lymph gland.

We subsequently analyzed the expression pattern of the MZ-specific markers *domeless-Gal4, UAS-2×EGFP* (*dome>GFP*), E-cadherin (Shg) and Patched (Ptc), which is a receptor in the Hh pathway. In contrast to wild-type larvae, the lymph glands of *jumu* heterozygotes third-instar larvae strongly expressed prohemocyte markers and presented a greatly enlarged MZ (*Figure 1O–Y*). However, the lymph glands of *jumu* double heterozygotes mutants did not show an obviously enlarged MZ compared with the wild-type (*Figure 1X,Y*). In addition, we also assessed ROS as a specific marker for MZ. ROS appear by the third instar, and increased levels of ROS can prime progenitor cells for differentiation (*Owusu-Ansah and Banerjee, 2009*). Consistent with the results of a previous study, ROS appeared in the MZ of the lymph gland of *w*$^{1118}$ mid-third-instar larvae and remained increased until late stages (*Figure 1Z*); however, ROS were rarely observed in *jumu* mutants during the entire third instar stage (*Figure 1AA-DD*). These results indicate that the loss of *jumu* prevents the differentiation of hematopoietic progenitors into plasmatocytes and crystal cells.

## Jumu negatively regulates the proliferation and ectopic expansion of Col$^+$ PSC cells

Previous studies have indicated that an increased PSC domain size can cause an increase in the MZ and a reduction of hemocyte differentiation (*Mandal et al., 2007*; *Pennetier et al., 2012*). To investigate whether an increase in the number of PSC cells could be observed in *jumu* mutants, we detected the expression of Antp, a specific marker of the PSC in the lymph gland. As predicted, the loss of *jumu* indeed led to substantial increases in the PSC size and cell number to different degrees (*Figure 2A–E,U*). In particular, the deletion mutants *jumu*$^{Df2.12}$ and *jumu*$^{Df3.4}$ showed strong increases in PSC cells and ectopic expansion of Antp; these ectopic PSC cells were scattered and often formed two or more cell clusters with higher accumulation in the MZ and CZ (*Figure 2C,D*). Moreover, the *jumu* double heterozygotes mutants showed more severe defects in PSC morphology, specifically the existence of PSC cells mostly in the secondary lobes (*Figure 2E*). In addition, the Hh signal from the PSC is required for the maintenance of prohemocytes (*Mandal et al., 2007*); thus, we subsequently detected Hh expression using the *hhF4-GFP* reporter gene. Similarly to the expression of Antp, *jumu* mutants also showed increased numbers of Hh$^+$ cells, and these were highly co-localized with Antp$^+$ PSC cells (*Figure 2F–G'*). It has been indicated that Col is also a PSC-specific marker and regulates Antp and Hh expression in the PSC in third-instar larvae (*Mandal et al., 2007*; *Pennetier et al., 2012*). Thus, we tested whether *jumu* affects the PSC by controlling the level of Col. Similar to Antp and Hh, increased Col was also observed in *jumu* mutant lymph glands, and increased numbers of *Col>GFP*$^+$ PSC cells were colocalized with Antp$^+$ cells (*Figure 2H–L*). Moreover, a reduced expression of Col can inhibit the phenotypes associated with *jumu* mutants: specifically clearly reduced numbers of both Col$^+$ and Antp$^+$ PSC cells accompanied by a reduction in the MZ size and an increased differentiation of plasmatocytes (*Figure 2M–U*).

To determine whether the ectopic PSC cells in *jumu* mutant lymph glands are migrated from the PSC or are de novo arising in the MZ and CZ, we first detected the morphology of the PSC at different larval stages. Expanded PSC cells were observed in *jumu* mutant L1 larvae and were amplified in

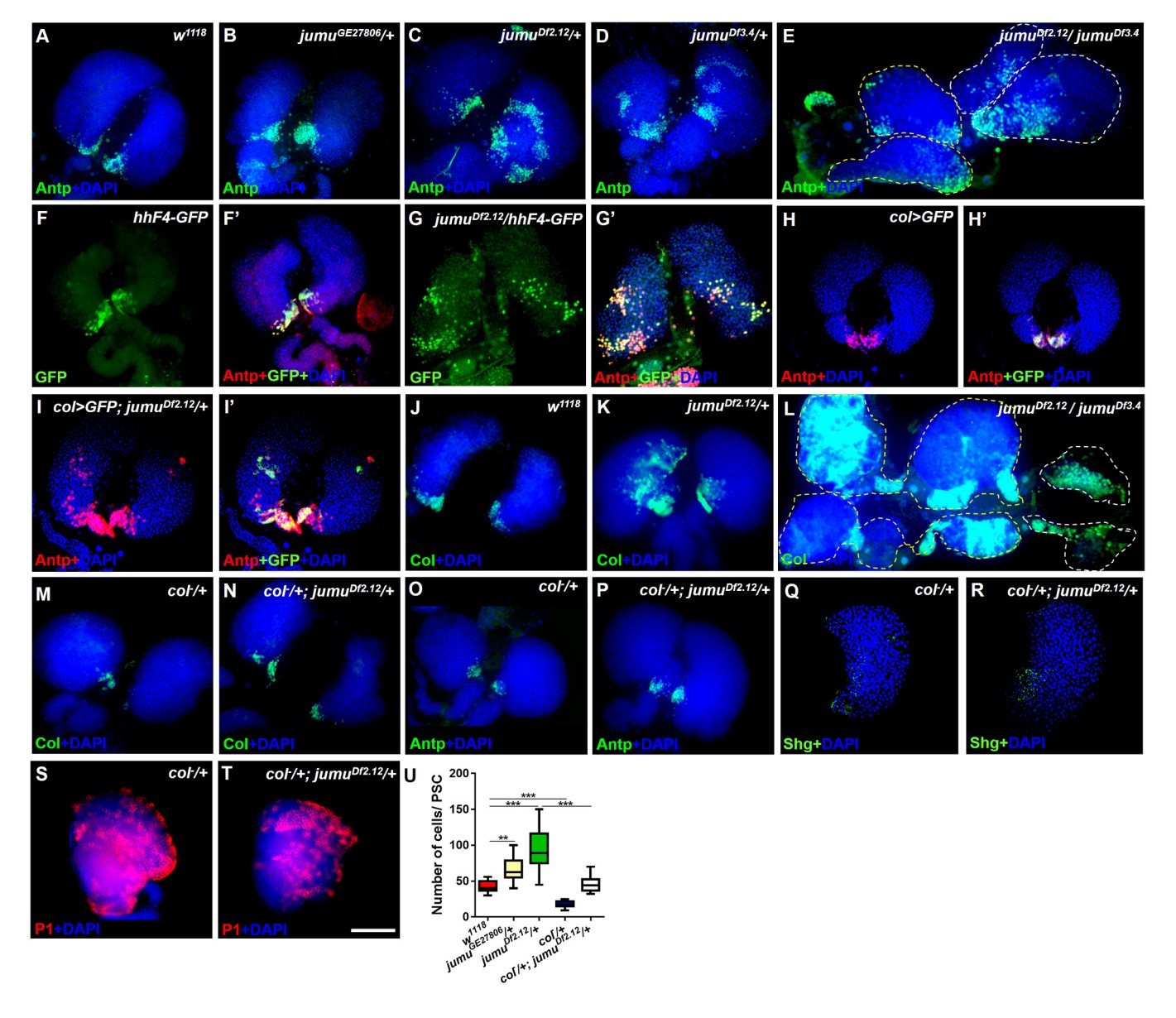

**Figure 2.** Loss of *jumu* causes expansion of the populations of Col+ PSC cells in the lymph gland. (A–E) Immunostaining against the PSC cell marker Antp (green) shows an increased number and ectopic expansion of PSC cells in *jumu*-mutant lymph glands. (F–G') Increased *hhF4f-GFP-positive* PSC cells (green) are located throughout the lymph glands of *jumuDf2.12* third-instar larvae, and these cells are co-localized with Antp (red). (H–I') Antp immunostaining (red) of the lymph glands of third-instar larvae expressing GFP under the control of *col-Gal4* (green) shows col>GFP ectopic expression and co-localization with Antp in *jumuDf2.12*. (J–L) Immunostaining against Col (green) shows ectopic expansion of Col+ PSC cells in *jumu*-mutant lymph glands. (M–P) Deficiency in the *col* levels (*col–/+*) reduces the numbers of both Col+ and Antp+ PSC cells in *jumuDf2.12*. (Q–T) Deficiency in the *col* levels (*col–/+*) reduces Shg expression (green) (Q, R) and increases the differentiation of plasmatocytes (red) (S, T) in *jumuDf2.12*. (U) Quantification of the number of PSC cells (Antp+ cells) per lymph gland lobe. The data are presented through a box-and-whisker representation. **p<0.01 and ***p<0.001 (one-way ANOVA). Dashed white and yellow lines outline the edges of the primary and secondary lobes, respectively, in E and L. Scale bars: 100 µm.

L2 and L3 larval stages (*Fgure 3A-L'*). In addition, we observed that the ectopically expanded Antp+ PSC cells did not co-localize with any *dome-GFP+* cells or *Hml-GFP+* cells in the MZ and CZ domains, particularly among the ectopic PSC cells adjacent to MZ or CZ cells during the early larval stage; in fact, we did not detect any intermediate cells co-expressing both Antp and GFP appeared (*Figure 3E–L'*). In summary, these results indicate that the ectopic PSC cells in *jumu* mutants are

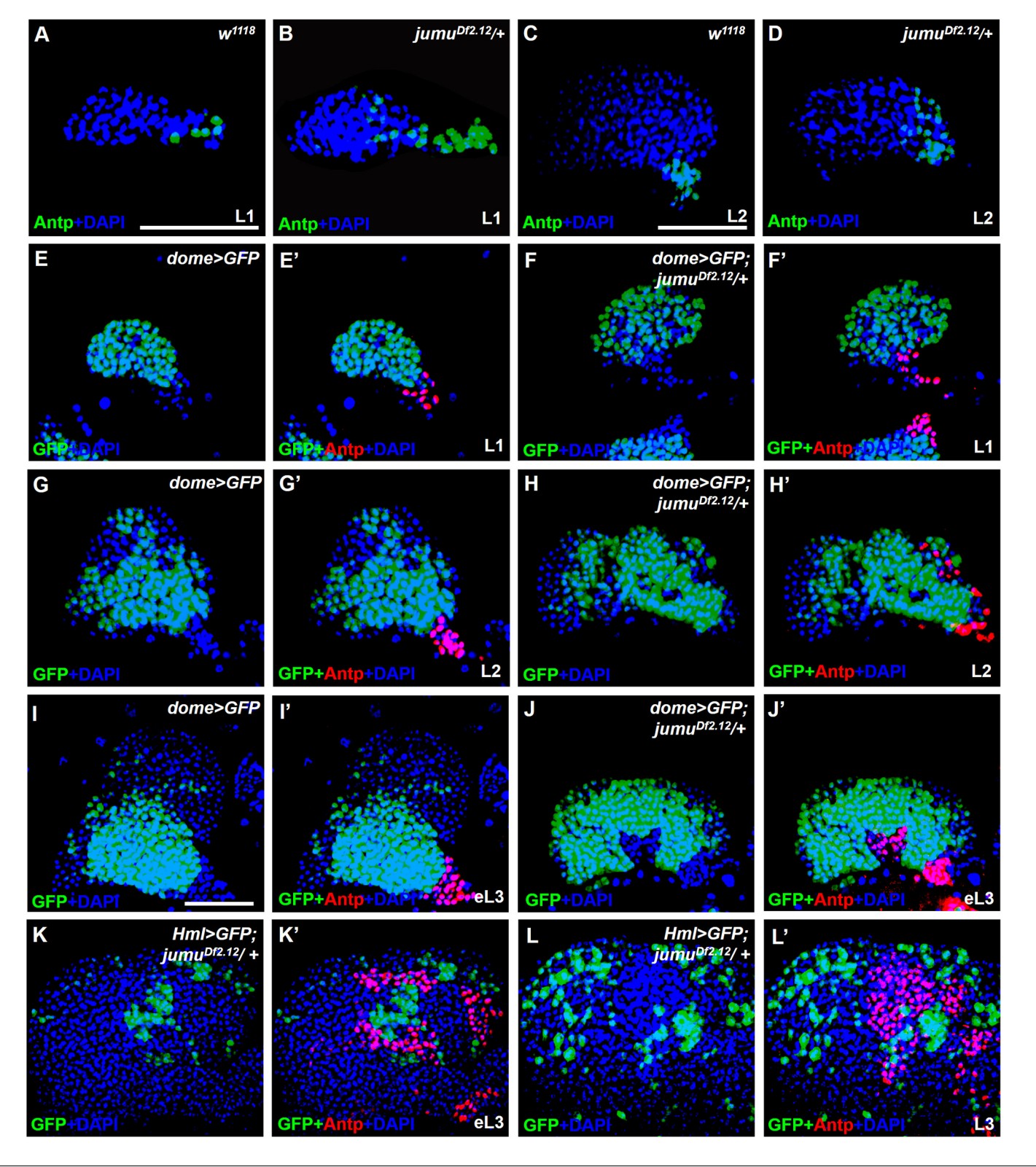

**Figure 3.** *jumu* is required for the prevention of ectopic PSC cells derived from the PSC domain during larval development. (A–D) Immunostaining against the PSC cell marker Antp (green) in L1 larvae (A, B) and L2 larvae (C, D). (E–J') The expanded Antp⁺ PSC cells (red) in *jumu*^Df2.12^ lymph glands show no co-localization with *dome-GFP*⁺ cells during the L1 larvae (E–F'), L2 larvae (G–H') and early L3 (I–J') stages. (K–L') The expanded Antp⁺ PSC cells (red) in *jumu*^Df2.12^ lymph glands show no co-localization with *Hml-GFP*⁺ cells during the early L3 (K, K') and L3 (L, L') stages. Scale bars: 50 μm.

derived from the PSC domain, and Jumu controls hematopoietic homeostasis by serving as a crucial negative regulator of Col[+] PSC cell proliferation and ectopic expansion.

## Jumu expression in prohemocytes is required for preventing the ectopic expansion of PSC cells

To further investigate where and how Jumu regulates the number and organization of PSC cells, we first identified the expression pattern of Jumu protein in the lymph gland at different larval stages through anti-Jumu antibody staining. In wild-type larvae, Jumu is expressed in the entire lymph gland and mainly located in the nuclei and presents markedly higher expression in the PSC than in the other domains during the L1 to L3 larval stage (*Figure 4A–F''*); however, the signal was nearly abolished in *jumu* double heterozygotes (*Figure 4E*). Moreover, we found that Jumu is also expressed in cardiomyocytes and is colocalized with Antp in some cells (*Figure 4A–A''*). A recent study suggested that Slit/Robo signaling from the cardiac tube is required to preserve the morphology and function of the PSC (*Morin-Poulard et al., 2016*). To determine whether the expansion of the PSC cells in *jumu* mutants is caused by the loss of *jumu* in cardiomyocytes, we used a heart-specific Gal4 driver (*Hand-Gal4*) to express *jumu* in *jumu*[Df2.12]. However, the PSC morphology defects detected in *jumu*[Df2.12] were not rescued by the overexpression of *jumu* (*Figure 4G,H and S*). We then used tissue-specific Gal4 lines to achieve a forced expression of *jumu* in the *jumu*[Df2.12] PSC niche (*Antp-Gal4*), MZ (*dome-Gal4, TepIV-Gal4*) and CZ (*Hml-Gal4*) and found that the overexpression of *jumu* in the PSC or CZ domain cannot rescue the ectopic expansion of PSC cells in *jumu*[Df2.12] (*Figure 4I,J,M,N and S*). However, the overexpression of *jumu* using *dome-Gal4* and *TepIV-Gal4* effectively inhibited the generation of expanded PSC cells in the MZ and CZ (*Figure 4K,L and S*; *Figure 5A,B and R*). Moreover, the overexpression of *jumu* in the entire lymph gland using the pan-lymph gland specific driver *Srp-Gal4* or the ubiquitous driver *Tub-Gal4*[ts] also rescued the PSC morphology defects detected in *jumu*[Df2.12] (*Figure 4O–S*). Thus, these results indicate that Jumu protein in prohemocytes might prevent the ectopic expansion of PSC cells through a non-cell-autonomous mechanism.

## The loss of *jumu* in the MZ non-cell-autonomously induces increases in dMyc in the PSC

To further determine whether *jumu* expression in the MZ non-cell-autonomously prevents the expansion of PSC cells, we knocked down or overexpressed *jumu* using the MZ-specific drivers *TepIV-Gal4* and *dome-Gal4* and assessed the resulting PSC morphology. Similar to *jumu* mutants, the *TepIV>-jumu RNAi* and *dome>jumu RNAi* mutants also showed scattered and increased numbers of PSC cells; however, the overexpression of *jumu* in MZ did not cause any defects in PSC morphology (*Figure 5A–D,R*; *Figure 6A–C,E*). We then used the pan-lymph gland specific driver *Srp-Gal4* and *e33C-Gal4* to knock down or overexpress *jumu* in the entire lymph gland and both *Srp-Gal4>jumu RNAi* and *e33C>jumu RNAi* displayed the scattered and increased numbers of PSC cells (*Figure 5E–I,R*). Because *e33C>UAS-jumu* was lethal during embryogenesis, we only detected the PSC of *Srp>UAS-jumu* and did not observe any changes in the morphology and cell number of the PSC domain (*Figure 5G,R*). These results strengthen the notion that Jumu protein located in the MZ mainly prevents the expansion and scattering of PSC cells toward the MZ and indirectly maintains the normal MZ development, but is not sufficient to initiate the inhibition of the proliferation of PSC.

Slit/Robo signaling controls the clustering and number of PSC cells by regulating DE-cadherin (Shg) and Cdc42 expression and dMyc activity, respectively (*Morin-Poulard et al., 2016*). Notably, a deficiency in Slit/Robo signaling resulted in defects in the PSC similar to those obtained in *jumu* mutants, and both Slit, which is the canonical ligand of *Drosophila* Robo, and Jumu are expressed in the dorsal vessel (*Morin-Poulard et al., 2016*). Thus, we first asked whether Jumu protein in cardiomyocytes controls Slit/Robo signaling and, in turn, regulates both proliferation and clustering of the PSC. However, the knockdown or overexpression of *jumu* under the control of *hand-Gal4* did not cause the abnormal PSC morphology (*Figure 6—figure supplement 1A–D*). This result suggests that Jumu does not directly regulate Slit/Robo signaling in the dorsal vessel. Moreover, the loss of Shg or the activation of Cdc42 in the PSC can induce the clustering of PSC cells (*Morin-Poulard et al., 2016*). Thus, we questioned whether Jumu protein in the MZ controls the PSC morphology by regulating the expression of Shg or the activation of Cdc42. Therefore, we first analyzed

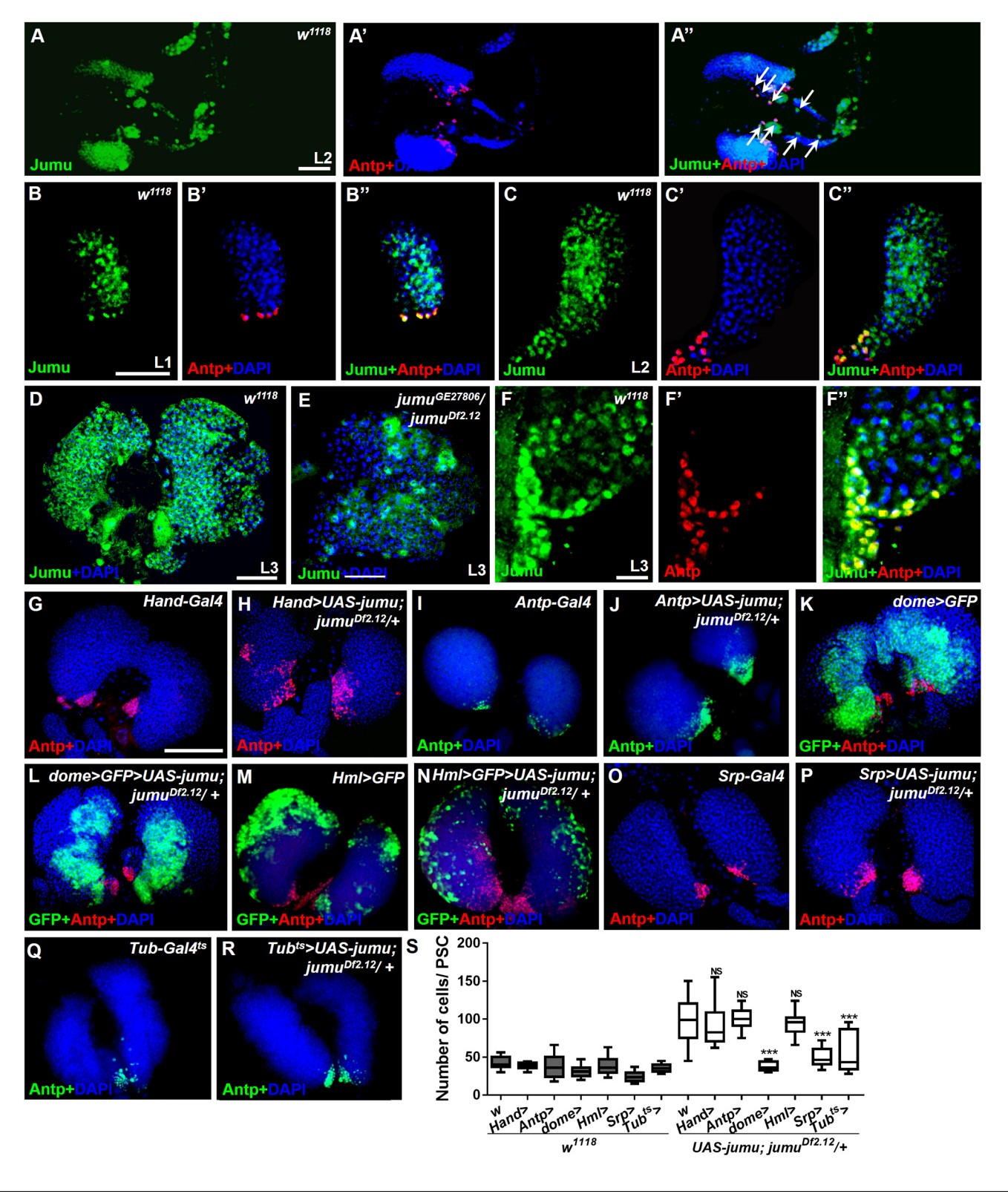

**Figure 4.** Expression of Jumu in prohemocytes is required for preventing the expansion of PSC cells. (A–D) Immunostaining against Jumu protein shows that Jumu (green) is expressed throughout the $w^{1118}$ lymph gland and shows co-localization with Antp (red) during the entire larval development process (A–D). Jumu is also expressed in cardiomyocytes and colocalizes with Antp in some cells (arrow) (A''). (E) The expression of Jumu (green) is obviously decreased in *jumu* double heterozygotes mutants. (F–F'') Jumu (green) is expressed at high levels in Antp+ PSC cells (red). (G–R) The

*Figure 4 continued on next page*

*Figure 4 continued*

overexpression of *jumu* using *Hand-Gal4* (heart specific), *Antp-Gal4* (PSC niche specific) and *Hml-Gal4* (CZ specific) in *jumu^Df2.12* lymph glands cannot inhibit the expansion of PSC cells (red) (**H, J, N**); however, the use of *dome-Gal4* (MZ specific), *Srp-Gal4* (entire lymph gland) or *Tub-Gal4^ts* (entire lymph gland) to overexpress *jumu* in *jumu^Df2.12* lymph glands effectively prevents the expansion of PSC cells (**L, P, R**). (**S**) Quantification of the number of PSC cells (Antp$^+$ cells). The data are presented in a box-and-whisker representation. NS, not significant, ***p<0.001 (Student's *t* test). Scale bars: 50 μm (**A–E**), 20 μm (**F–F''**), and 100 μm (**G–R**).

Shg expression in the PSC and found that the expression levels of Shg were not reduced in the PSC of *jumu^Df2.12* compared with the wild-type (*Figure 5—figure supplement 1A–B''*), suggesting that the role of Jumu in the PSC morphology is independent of Shg. Moreover, the overexpression of the dominant negative form of Cdc42 (*cdc42-DN*) in the PSC of *jumu^Df2.12* did not rescue the increased number and clustering of the PSC cells (*Figure 5—figure supplement 1C–E*). Slit/Robo signaling controls PSC cell proliferation via *dmyc*, and promotes *dmyc* transcription through the negative regulation of the HSPG (heparin sulfate proteoglycan) Dally-like (Dlp) and BMP/Dpp signaling pathways (*Morin-Poulard et al., 2016*). We subsequently asked whether Jumu also controls the PSC cell number by regulating dMyc expression. The use of *TepIV-Gal4* to knock down or overexpress *jumu* did not change the expression levels of dMyc in the MZ and CZ; however, we observed increased dMyc expression in the PSC cells of *TepIV>jumu RNAi* mutants (*Figure 5J–L', S*). In addition, we detected dMyc in the lymph gland of *e33C>jumu RNAi* and *jumu* mutant larvae. The loss of *jumu* throughout the lymph gland did not cause an obvious reduction of dMyc in the MZ and CZ but caused an observable increase in the dMyc levels in PSC cells (*Figure 5M–Q', S*). Moreover, the knockdown of *dmyc* in the PSC of *jumu^Df2.12* effectively suppressed the scattering of and the increase in the number of PSC cells (*Figure 5T–V*). These results indicate that Jumu protein in prohemocytes might be required to prevent the overexpression of dMyc in the neighboring PSC cells in a non-cell-autonomous manner and that it accordingly inhibits the overproliferation and ectopic presence of PSC cells in an indirect manner. The loss of BMP/Dpp activity leads to increased numbers of PSC cells through the up-regulation of the expression of dMyc, and Dpp signaling is regulated directly by Dlp (*Pennetier et al., 2012*; *Morin-Poulard et al., 2016*). Thus, we subsequently detected whether Jumu controls dMyc expression by regulating Dlp and Dpp signaling. However, the normal expression of Dlp and phosphorylated Mad (P-Mad), which targets the activity of the BMP/Dpp signaling pathway, were detected in the PSC cells of *jumu^Df2.12* (*Figure 5—figure supplement 1F–I''*). Taken together, these results suggest that Jumu in MZ might control the morphology of the PSC by indirectly regulating dMyc expression in PSC cells through a Slit/Robo and BMP/Dpp-independent mechanism.

## Jumu cell-autonomously controls the maintenance of blood cell progenitors by regulating the low levels of Col in the MZ

Because Jumu is expressed throughout the lymph gland, we investigated the role of *jumu* in the various domains of the lymph gland. We knocked down or over expressed *jumu* under the control of a driver specific to differentiating cells (*Hml-Gal4*), but this had no effect on the size of the PSC niche or on prohemocyte differentiation (*Figure 6—figure supplement 1E–L*). To determine whether Jumu is involved in prohemocyte differentiation in a cell-autonomous manner in addition to its non-cell-autonomous role in the regulation of PSC morphology, we used the *dome-Gal4* driver to knock down or overexpress *jumu* in the MZ. Although an expansion of the neighboring PSC population was observed (*Figure 6A,B and E*), the knockdown of *jumu* in prohemocytes did not obviously reduce plasmatocyte or crystal cell differentiation (*Figure 6F,G,J,K,L and O*). However, the overexpression of *jumu* in the MZ resulted in massive differentiation of plasmatocytes and crystal cells and a reduction in hematopoietic progenitors (*Figure 6H,J,M and O*). Because *dome-Gal4* is also expressed in the brain, and some signaling pathways derived from the brain also control the differentiation of prohemocytes (*Shim et al., 2013*), we examined the possible involvement of the brain by overexpressing or knocking down *jumu* using the brain-specific driver *elav-Gal4*, but this did not result in an abnormal differentiation of prohemocytes (*Figure 6—figure supplement 1M–P*). Because the forced expression of *jumu* in the MZ was not sufficient to reduce the number of PSC cells (*Figure 6C,E*), the overdifferentiation of prohemocytes could not be indirectly caused by the

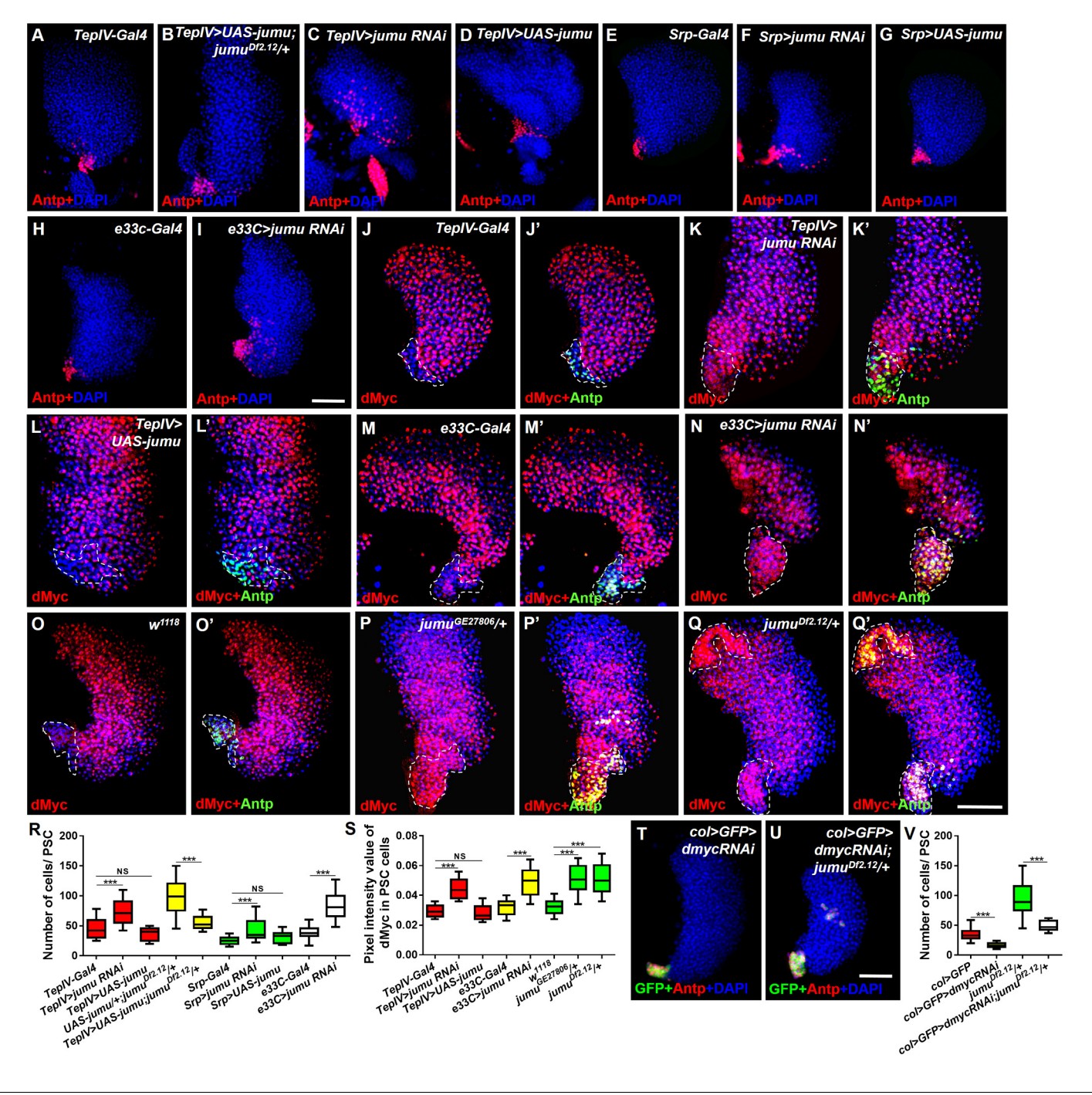

**Figure 5.** Jumu in prohemocytes non-cell-autonomously regulates the expression of dMyc in the PSC. (**A–I**) The knockdown of *jumu* in prohemocytes (*TepIV>jumu RNAi*) (**C**) or in the entire lymph gland (*Srp>jumu RNAi, e33C>jumu RNAi*) (**F, I**) causes the expansion of Antp⁺ PSC cells (red), and the overexpression of *jumu* in prohemocytes (*TepIV>UAS-jumu*) or in the entire lymph gland (*Srp>UAS-jumu*) does not affect the number of PSC cells (**D, G**). (**J–L'**) The knockdown of *jumu* in prohemocytes (*TepIV>jumu RNAi*) causes increased levels of dMyc (red) in PSC cells (green) (**K, K'**), but the dMyc level is not changed in *TepIV>UAS-jumu* (**L, L'**). (**M–Q'**) The expression of dMyc (red) is substantially increased in PSC cells (green) of the *e33C>jumu RNAi* (**N, N'**), *jumu^GE27806/+* (**P, P'**) and *jumu^Df2.12* mutants (**Q, Q'**). (**R**) Quantification of the number of PSC cells (Antp⁺ cells). (**S**) Quantification of the pixel intensity of dMyc (integrated density of the dMyc signal in the PSC divided by the PSC area). (**T–V**) The use of *col-Gal4* to knock down *dmyc* can rescue the expanded PSC cells in *jumu^Df2.12*. All of the quantification data are presented as box-and-whisker representations. NS, not significant, ***p<0.001 (Student's *t* test). Dashed white lines outline the PSC in J-Q'. Scale bars: 50 μm.

*Figure 5 continued on next page*

*Figure 5 continued*

The following figure supplement is available for figure 5:

**Figure supplement 1.** Jumu protein in the MZ controls PSC morphology through a Slit/Robo- and BMP/Dpp-independent mechanism.

reduction in the number of neighboring PSC cells. Hh signaling from the PSC is required for the maintenance of hematopoietic progenitors (*Mandal et al., 2007*). Thus, we subsequently detected Hh expression in the PSC using the HhF4-GFP reporter gene. However, the overexpression of *jumu* in the MZ did not reduce the HhF4-GFP expression levels in the PSC, and the knockdown of *jumu* in the MZ also did not increase the HhF4-GFP expression levels in the PSC (*Figure 6—figure supplement 2*). Based on these results, *jumu* might control the differentiation of prohemocytes through other cell-autonomous mechanisms.

To further confirm whether Jumu cell-autonomously affects prohemocyte differentiation, we knocked down or overexpressed *jumu* in lymph gland-specific clonal populations of hemocytes using *HHLT-Gal4* (*Hand-Hml lineage-traced Gal4*). A proportion of P1$^+$ differentiated plasmatocytes were colocalized with the wild-type clones (GFP$^+$) generated by *HHLT-Gal4* (*Figure 6—figure supplement 3A–A''*). However, most P1$^+$ differentiated plasmatocytes did not overlap with clonal cells lacking *jumu* (GFP$^+$) (*Figure 6—figure supplement 3B–B''*), suggesting that the loss of *jumu* in clonal cells can inhibit hemocyte differentiation cell-autonomously. Conversely, most P1$^+$ cells were colocalized with clonal cells overexpressing *jumu* (GFP$^+$) (*Figure 6—figure supplement 3C–C''*), suggesting that the overexpression of *jumu* in clonal cells can promote hemocyte differentiation in a cell-autonomous manner. Altogether, the HHLT clonal analyses indicate the cell-autonomous requirement of Jumu in controlling lymph gland hemocyte differentiation.

A recent study suggested that in addition to its expression in the PSC, Col is also expressed at low levels in prohemocytes and is required for preventing the differentiation of prohemocytes (*Benmimoun et al., 2015*). The above-described results show that the loss of *col* in the lymph gland of the *jumu* mutants also rescued the increased MZ size and the reduced differentiation of plasmatocyte phenotypes in addition to inhibiting the expansion of PSC cells (*Figure 2M–T*). Thus, we evaluated whether Jumu could cell-autonomously inhibit the expression of Col in prohemocytes in addition to preventing the ectopic expansion of Col$^+$ PSC cells. The intensity of the Col signal in the MZ was quantified. As predicted, the expression of Col was clearly increased in the *dome>jumu RNAi* lymph gland and significantly reduced in the *dome>UAS-jumu* lymph gland (*Figure 6P–R', V*). Similarly, *jumu* mutants also displayed an increased Col expression signal in the MZ (*Figure 6S–V*). A previous study suggested that Col regulates the expression of (Dlp) in the PSC (*Pennetier et al., 2012*). We found that Dlp is also expressed in the MZ (*Figure 6W,W'*), and the overexpression of *col* in the MZ enhanced the production of Dlp protein, an effect that was accompanied by the expansion of the MZ (*Figure 6X,X 'and AA*). Consistent with this phenotype, we found increased and reduced levels of Dlp expression in the MZ of *dome>jumu RNAi* and *dome>UAS-jumu* mutants, respectively (*Figure 6Y–AA*). In addition, the *dome*-driven expression of *col* was sufficient to suppress the plasmatocyte and crystal cell differentiation phenotypes associated with *jumu* overexpression in prohemocytes (*Figure 6I,J,N and O*). These results demonstrate that Jumu cell-autonomously promotes the differentiation of prohemocytes by inhibiting the expression of Col in the MZ. Moreover, the overexpression of Col in prohemocytes did not induce the ectopic expansion of PSC cells (*Figure 6D,E*); thus, the expanded PSC cell phenotype caused by *jumu* loss-of-function is independent of the increased expression of Col in the MZ.

## Jumu cell-autonomously controls the size of the PSC by regulating the expression of dMyc

To test whether *jumu* is also cell-autonomously required in the PSC, we used a *col-Gal4* driver to knock down and overexpress *jumu* in the PSC. The expression of dome-MESO, a reporter of JAK-STAT signaling, was used to identify prohemocytes. In contrast to knockdown in the MZ or the entire lymph gland, the knockdown of *jumu* in the PSC (*col>jumu RNAi*) led to a reduction in the number of PSC cells and the overdifferentiation of plasmatocytes at the expense of the MZ; consistently, the overexpression of *jumu* in the PSC (*col>UAS-jumu, Antp>UAS-jumu*) substantially increased the

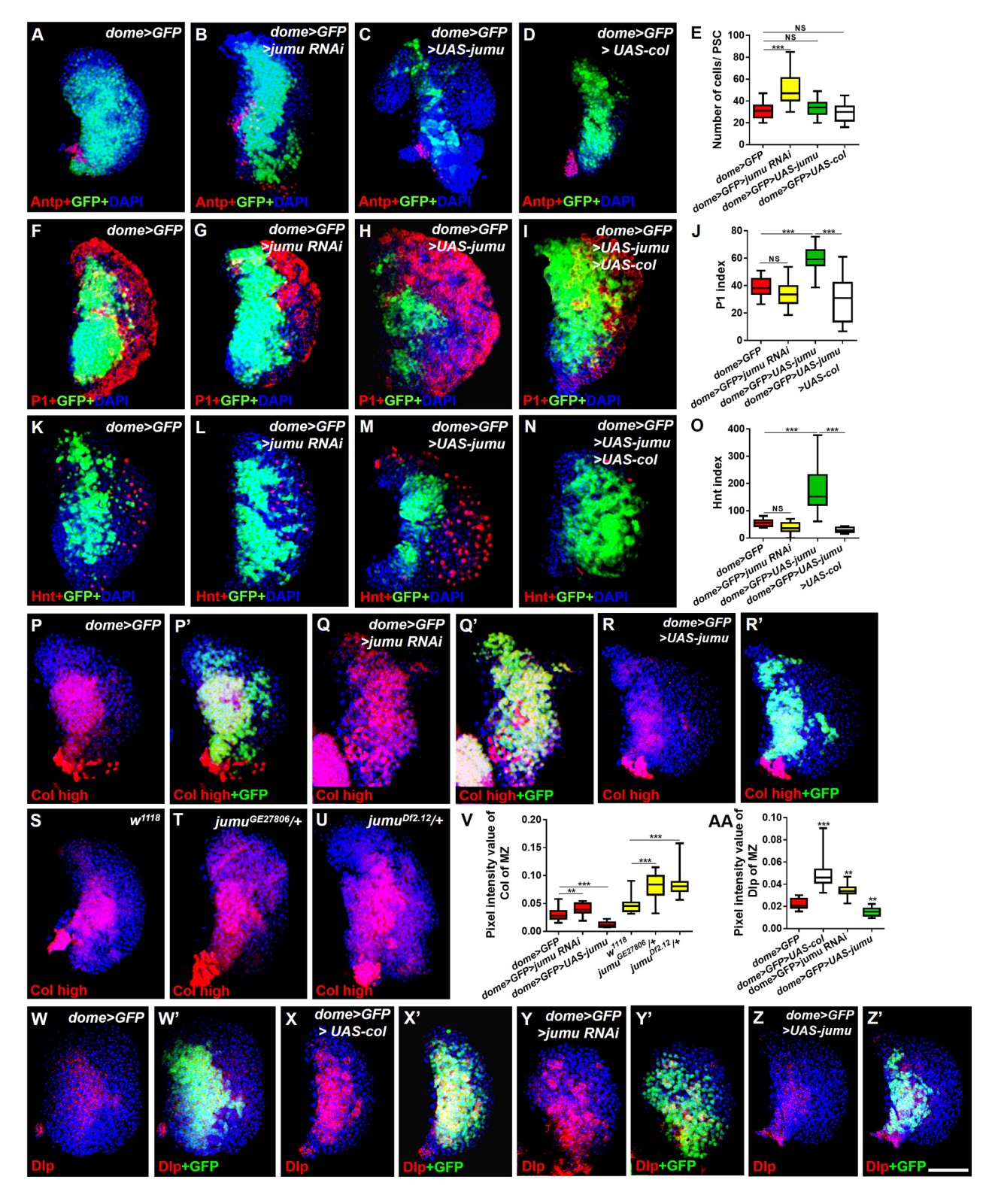

**Figure 6.** Jumu cell-autonomously regulates the differentiation of blood cell progenitors by maintaining low expression levels of Col of the MZ. (**A–D**) Immunostaining against Antp shows that the knockdown of *jumu* in the MZ causes ectopic and increased PSC cells (**B**), but the overexpression of *jumu* or *col* in the MZ does not change the number of PSC cells (**C, D**). (**E**) Quantification of the number of PSC cells (Antp⁺ cells). The data are shown through a box-and-whisker representation. NS, not significant, \*\*\*p<0.001 (Student's *t* test). (**F–I, K–N**) The knockdown of *jumu* in prohemocytes does

*Figure 6 continued on next page*

*Figure 6 continued*

not cause an obvious reduction in plasmatocyte and crystal cell differentiation compared with the control (G, L), whereas the forced expression of *jumu* in prohemocytes causes increases in plasmatocyte and crystal cell differentiation (H, M). The overexpression of *col* suppresses the increases in plasmatocyte and crystal cell numbers in *dome>GFP>UAS-jumu* (I, N). (J, O) Quantification of plasmatocyte and crystal cell indexes in primary lobes. The data are presented in a box-and-whisker representation. NS, not significant, ***p<0.001 (one-way ANOVA). (P–U) High-exposure visualization of Col immunostaining (red) shows that compared with the controls, the Col levels of the MZ are increased in *dome>GFP>jumu RNAi* (Q, Q') and in heterozygous *jumu* mutants (T, U) but are reduced in *dome>GFP>UAS-jumu* (R, R'). (W–Z') Dlp (red) is expressed in the MZ of *dome>GFP* and is increased in *dome>GFP>UAS-col* (X, X') and *dome>GFP>jumu RNAi* (Y, Y'); however, Dlp expression is reduced in *dome>GFP>UAS-jumu* (Z, Z'). (V, AA) Quantification of the pixel intensity of Col or Dlp (integrated density of Col or Dlp signal in the MZ divided by the area of the primary lobe). The data are presented in a box-and-whisker representation. **p<0.01, ***p<0.001 (Student's *t* test). Scale bars: 50 μm.

The following figure supplements are available for figure 6:

**Figure supplement 1.** Jumu protein in the heart, the CZ of the lymph gland and the brain does not regulate the number of PSC cells or prohemocyte differentiation.

**Figure supplement 2.** Changes in the expression levels of *jumu* in the MZ do not affect the expression of Hh in the PSC.

**Figure supplement 3.** Jumu cell-autonomously regulates the differentiation of blood cell progenitors and PSC proliferation.

number of PSC cells and resulted in an expansion of the MZ at the expense of hemocyte differentiation (*Figure 7A–L*; *Figure 7—figure supplement 1A–C*). Moreover, we also observed the similar phenotypes by clonally knocking down or overexpressing *jumu* in the PSC using *HHLT-Gal4* (*Figure 6—figure supplement 3D–F*). These results suggest that Jumu cell-autonomously controls the size and cell number of the PSC. A previous study suggested that a reduction in the Col levels in the PSC can cause an increase in the number of PSC cells by down-regulating Dlp and Hh expression (*Pennetier et al., 2012*). The above-described results indicate that *jumu* negatively regulates the expression of Col in prohemocytes. Thus, we first asked whether the overexpression of *jumu* also reduces the expression of Col in the PSC, thereby indirectly increasing the number of PSC cells. However, although the number of Col$^+$ PSC cells was significantly increased, the average pixel intensity of the Col signal per PSC cell was not reduced in the *Antp>UAS-jumu* mutant (*Figure 7—figure supplement 1D–F*). In addition, the overexpression of *jumu* in the PSC did not result in a loss of Hh and Dlp expression, which are directly regulated by Col (*Figure 7—figure supplement 1G–K'*). Dpp signaling is regulated directly by Dlp and indirectly by Col (*Pennetier et al., 2012*). We subsequently detected whether Jumu protein in the PSC controls the proliferation of PSC cells by suppressing the activity of the Dpp signaling pathway. The normal expression level of P-Mad could be detected in the *Antp>UAS-jumu* mutant (*Figure 7—figure supplement 1L–M"*). These results indicate that *jumu* cell-autonomously controls the size of the PSC niche through a Col-independent and Dpp-independent mechanism.

Recent studies have suggested that some genes and signaling pathways control the PSC cell number and size by regulating dMyc expression in the PSC (*Pennetier et al., 2012*; *Sinenko et al., 2009*; *Benmimoun et al., 2012*; *Tokusumi et al., 2015*). Moreover, the above results suggest that Jumu protein in the MZ non-cell-autonomously prevents an increase in dMyc expression in the PSC. In addition, the knockdown of *dmyc* or *jumu* in the PSC resulted in a similar phenotype of fewer PSC cells, and correspondingly, the overexpression of *dmyc* or *jumu* in the PSC also caused increased numbers of PSC cells. Thus, we hypothesized that the existence of a regulatory relationship between Jumu and dMyc. Therefore, we first detected the expression of dMyc in *jumu* loss- and gain-of-function lymph glands. Surprisingly, a reduced level of dMyc was observed in the PSC of *col>jumu* RNAi lymph glands, whereas an increased level of dMyc was observed in *col>UAS-jumu* lymph glands (*Figure 7M–O', R*). However, the knockdown of *dmyc* in the PSC did not result in an obvious absence of Jumu expression in the PSC (*Figure 7P–Q", S*). Additionally, the knockdown of *dmyc* in PSC cells can suppress the increase in PSC cells caused by the overexpression of *jumu* (*Figure 7T–V*). These results indicate that Jumu is a direct or indirect positive regulator of dMyc expression in the PSC and promotes the overproliferation of PSC cells.

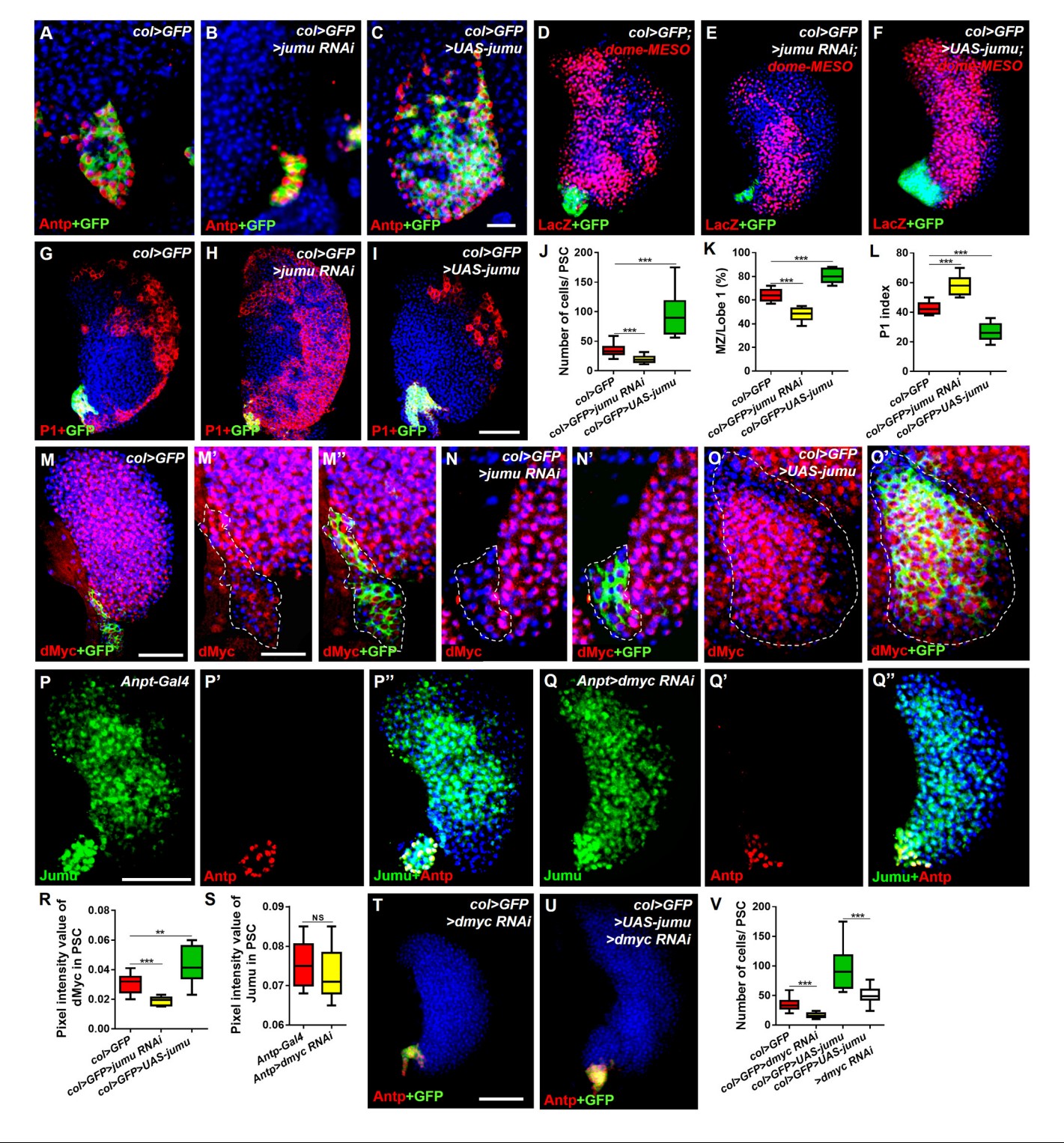

**Figure 7.** Jumu cell-autonomously controls the size of the PSC by regulating the expression of dMyc and indirectly affects prohemocyte differentiation. (A–C) Immunostaining against Antp (red) shows that *jumu* RNAi in the PSC (*col>GFP>jumu RNAi*) causes a reduction in the number of PSC cells (B), but the overexpression of *jumu* in the PSC (*col>GFP>UAS-jumu*) can substantially increase the size and cell number of the PSC niche (anti-Antp, red) (C). (D–F) The use of *dome-MESO* (LacZ) (red), which serves as a marker of prohemocytes, shows a reduction of the MZ in *col>GFP>jumu RNAi* (E) and an expansion of the MZ in *col>GFP>UAS-jumu* (F). (G–I) Plasmatocyte differentiation (anti-P1, red) is increased in *col>GFP>jumu RNAi* and reduced in *col>GFP>UAS-jumu*. (J) Quantification of the number of PSC cells (Antp⁺ cells). (K) Quantification of the proportion of the primary lobes occupied by

*Figure 7 continued on next page*

*Figure 7 continued*

prohemocytes (*dome-MESO*⁺ MZ area divided by the primary lobe area). (L) Quantification of the plasmatocyte index in primary lobes. (M–O')
Compared with the control (*col>GFP*), the expression of dMyc (red) is reduced in the PSC of *col>GFP>jumu RNAi* (N, N') and significantly increased in
*col>GFP>UAS-jumu* (O, O'). Dashed white lines outline the PSC. (P–Q'') The levels of Jumu (green) are not decreased in the PSC cells (red) of
*Antp>dmyc RNAi*. (R, S) Quantification of the pixel intensity of dMyc or Jumu (integrated density of dMyc or Jumu signal in the PSC divided by the PSC
area). (T, U) The knockdown of *dmyc* in the PSC (*col>GFP>dmyc RNAi*) reduces the number of PSC cells (anti-Antp, red) (T) and suppresses the increase
in PSC cells observed in *col>GFP>UAS-jumu* (U). (V) Quantification of the number of PSC cells (Antp⁺ cells). The quantification data are presented as
box-and-whisker representations. NS, not significant. **$p<0.01$ and ***$p<0.001$ (Student's *t* test) in J-L, R and S; ***$p<0.001$ (one-way ANOVA) in V.
Scale bars: 20 µm (A–C, M'–O') and 50 µm (D–I, M, P–Q'', T, U).

The following figure supplements are available for figure 7:

**Figure supplement 1.** Jumu controls the size of the PSC through Col- and Dpp-independent mechanisms.

**Figure supplement 2.** Expression pattern of dMyc in the lymph gland during larval development.

**Figure supplement 3.** Spatial-temporal regulation of Jumu on PSC morphology during larval development.

The above results show that Jumu in the MZ does not cell-autonomously regulate dMyc expression in the MZ (*Figure 5J–L'*). Similarly, the loss or overexpression of *jumu* in the CZ did not cause obvious changes in the dMyc levels (data not shown). These results suggest that the role of Jumu in the regulation of dMyc expression is specifically restricted to the PSC of the lymph gland. To further address whether Jumu can directly regulate *dmyc* transcription, we subsequently detected the protein and transcription levels of the *dmyc* gene in circulating hemocytes and found that the overexpression of *jumu* in circulating hemocytes led to increases in the dMyc protein level and the transcription of the *dmyc* gene (*Figure 8A,B*). We then analyzed the 2000 bp promoter sequence of the *dmyc* gene using Ensembl, a web system for predicting promoters, and found 23 putative FKH transcription factor- binding sites on the promoter sequence using JASPAR, a database of transcription factor binding profiles (*Figure 8C,D*). Because Jumu contains a conserved FKH DNA-binding domain, we then examined whether Jumu regulates the expression of *dmyc* at the transcriptional level by directly binding to its promoter via the FKH domain in vivo through a chromatin immunoprecipitation (ChIP) assay. We designed specific primers for possible FKH domain-binding sites in the *dmyc* promoter (*Figure 8D*). Total lysates from S2 cells stably expressing Flag-Jumu were used as an input positive control, along with immunoprecipitation with IgG (as a negative control) or antibodies against Flag, and the immunoprecipitated chromatin fragments were detected using PCR. As shown in *Figure 8E*, using anti-Flag antibodies compared with IgG control, there was no obvious positive signal at the *dmyc* promoter region of −1993 bp to −105 bp, but a specific enrichment of the *dmyc* gene region of −46 bp to +235 bp was observed. This fragment contains three putative FKH-binding sites, one inside the promoter region and two inside the 5'UTR (*Figure 8F*). To further determine the binding site recognized by Jumu in the *dmyc* promoter, we mutated each of the three putative FKH-binding sites at two positions crucial for the efficiency of FKH binding (*Figure 8G*). The Fragment from −643 to +467 of the *dmyc* promoter was cloned into the luciferase reporter vector pGL3, and a reporter assay showed that the co-transfection of S2 cells with a *jumu* expression construct and pGL3-*dmyc* (WT-643/+467) resulted in a nearly two-fold induction of reporter gene activity compared with the control. The disruption of the Site 1 (Mut1) led to an almost three-fold reduction of reporter gene activity compared with WT-643/+467, whereas the mutation of the Site 2 (Mut2) and Site 3 (Mut3) did not affect the reporter gene activity. Taken together, these results suggest that Jumu can be endogenously recruited to the conserved FKH-binding sites located in −27/−17 of the *dmyc* promoter via its FKH domain, and further strengthen the possibility that Jumu positively upregulates *dmyc* expression by directly binding to the *dmyc* promoter.

The above results show that the knockdown of *jumu* in the MZ and PSC has opposing effects on the expression of dMyc and PSC morphology. We subsequently analyzed the spatial-temporal regulation of Jumu on PSC development during larval development. First, the expression pattern of dMyc during larval development was detected. dMyc is expressed at high levels in MZ and CZ cells but at low levels in PSC cells, and its expression pattern remains the same at all stages of larval

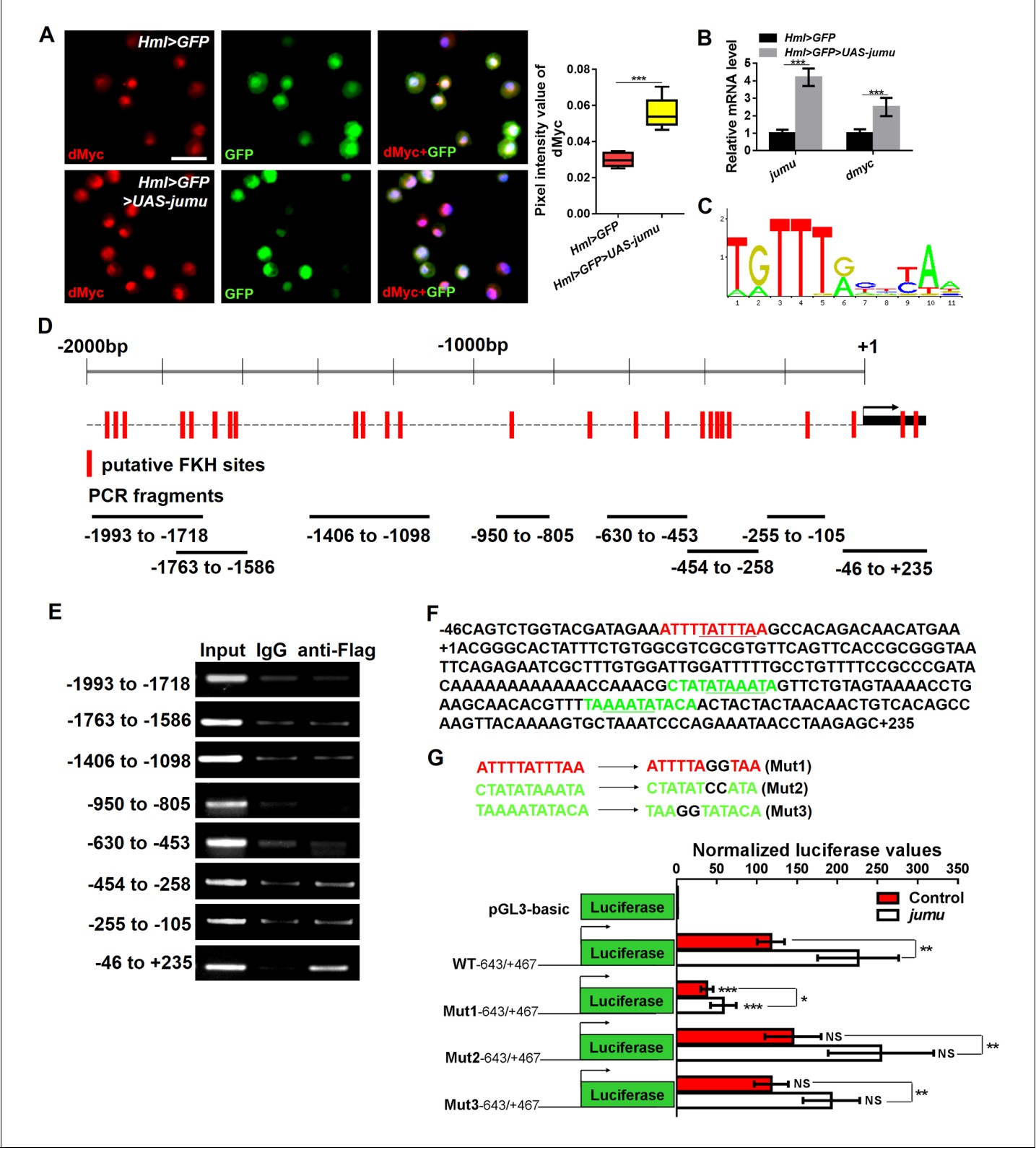

**Figure 8.** Jumu regulates the expression of *dmyc* by binding to its FKH-binding site. (**A, B**) The overexpression of *jumu* in circulating hemocytes (*Hml>GFP>UAS-jumu*) up-regulates *dmyc* expression at the translation (**A**) and transcription levels (**B**). ***p<0.001 (Student's *t* test). Scale bars: 20 μm in A. (**C**) Optimal FKH DNA-binding consensus motif obtained from the JASPAR database. (**D**) Schematic of the *dmyc* gene promoter locus showing putative FKH-binding sites (red lines) and DNA fragments from ChIP PCR amplification (black lines). (**E**) S2 cells stably expressing Flag-jumu were used

*Figure 8 continued on next page*

*Figure 8 continued*

for ChIP experiments to screen for the presence of endogenous Jumu on the *dmyc* promoter region. The enrichment of target DNA fragments relative to IgG is shown on an agarose gel. ChIP-PCR demonstrates that Jumu can bind to the *dmyc* gene in the region from −46 bp to +235 bp. (**F**) The analysis of the *dmyc* gene region from −46 bp to +235 bp contains one putative FKH site (red) inside *dmyc* promoter region and two putative FKH sites (green) inside the 5'UTR of *dmyc*. (**G**) Three putative FKH-binding sites (in −46 bp to +235 bp of the *dmyc* gene region) were mutagenized. The quantification of the results from a dual-luciferase reporter assay of S2 cells transfected with pMK33-*jumu* (white bars) or empty pMK33 vector (red bars) shows the activation of the wild-type (WT) *dmyc* promoter and that the mutagenesis of the Mut1 site, but not the Mut2 or Mut3 sites, prevents activation of the *dmyc* promoter. The normalized luciferase values are the ratios of firefly-to-Renilla luciferase. The data are presented as the means ± S.E.M. NS, not significant, *p<0.1, **p<0.01, ***p<0.001 (Student's *t* test).

development (**Figure 7—figure supplement 2**). Similarly, the expression pattern of Jumu does not change during larval development (**Figure 4A–F''**), suggesting that Jumu protein in the PSC might be responsible for the maintenance of dMyc expression. We then used a temporal and regional gene expression targeting system (Gal80ts/Gal4 expression system) (**McGuire et al., 2003**) to knock down *jumu* in the PSC at different larval stages (**Figure 7—figure supplement 3M**). Similar to the phenotype observed after the knockdown of *jumu* in the PSC from embryogenesis, the knockdown of *jumu* in the PSC from L1 larval stage or L2 stage also caused the reductions in the PSC cell number to different degrees; in fact, the earlier knock down of *jumu* resulted in a greater reduction in the PSC cell number (**Figure 7—figure supplement 3A–F,N**). This result indicates a continuous role for Jumu in PSC cell proliferation during larval development. The above results show that the loss of *jumu* in the MZ is responsible for the ectopic PSC cells detected in the early larval stages (**Figure 3A–D**). We then used the Gal80ts/Gal4 expression system to knock down *jumu* in the MZ at different larval stages (**Figure 7—figure supplement 3M**). The loss of *jumu* in the MZ from L1 larval stage or L2 stage also caused the ectopic expansion of PSC cells, and an earlier knockdown of *jumu* resulted in an increased ectopic expansion of PSC cells (**Figure 7—figure supplement 3G–L,O**). This result further confirms that the expression of Jumu in the MZ prevents the expansion of PSC cell toward the MZ throughout larval development. In conclusion, the expression of Jumu in the MZ and PSC simultaneously maintains the normal development of the PSC. Moreover, changes in the *jumu* expression levels in the entire lymph gland result in defects in PSC morphology similar to those detected in response to changes in the *jumu* levels in the MZ but opposite to those caused by changes in the *jumu* levels in the PSC, indicating that the expression of Jumu in the MZ might play a more prominent role in regulating PSC morphology when *jumu* is lost or overexpressed in both the PSC and the MZ.

## The loss of *jumu* in the entire lymph gland leads to the generation of lamellocytes through the activation of Toll signaling

The above-described results suggest that the severe loss of *jumu* can induce an increase in lamellocytes (**Figure 1K–N**). We subsequently investigated where and how Jumu controls the generation of lamellocytes. The knockdown of *jumu* in different zones of the lymph gland did not cause obvious increases in the lamellocytes; in fact, only less than 20% of the lymph gland showed a few lamellocytes (n > 30) (**Figure 9—figure supplement 1A–C**). However, using *e33C-Gal4* to knock down *jumu* in the entire lymph gland led to the generation of numerous lamellocytes in more than 50% of the lymph gland (n = 32), and similar to *jumu* mutant, *e33C>jumu RNAi* also showed reduced numbers of crystal cells (**Figure 9A–E**). Similarly, more than 60% of lymph gland of the *Srp> jumu RNAi* lymph gland also showed obvious increased numbers of lamellocytes (n = 36) (**Figure 9F,G**). Our previous study demonstrated that the overexpression of *jumu* in both hemocytes and the fat body induces the deposition of hemocytes and melanotic nodules through the activation of Toll signaling (**Zhang et al., 2016**). The constitutive activation of Toll signaling in hemocytes or the fat body can lead to the differentiation of lamellocytes (**Schmid et al., 2014**). Moreover, a recent study show that the Toll signaling transcription factors Dorsal and Dorsal-related immunity factor (Dif) are expressed in the lymph gland and that the ectopic expression of both factors in the PSC can cause the generation of lamellocytes in the lymph gland (**Gueguen et al., 2013**). Thus, we asked whether the lamellocytes caused by the loss of *jumu* in the lymph gland are linked to Toll signaling. The expression and localization of Dorsal and Dif were detected via antibody staining, and interestingly, we found that

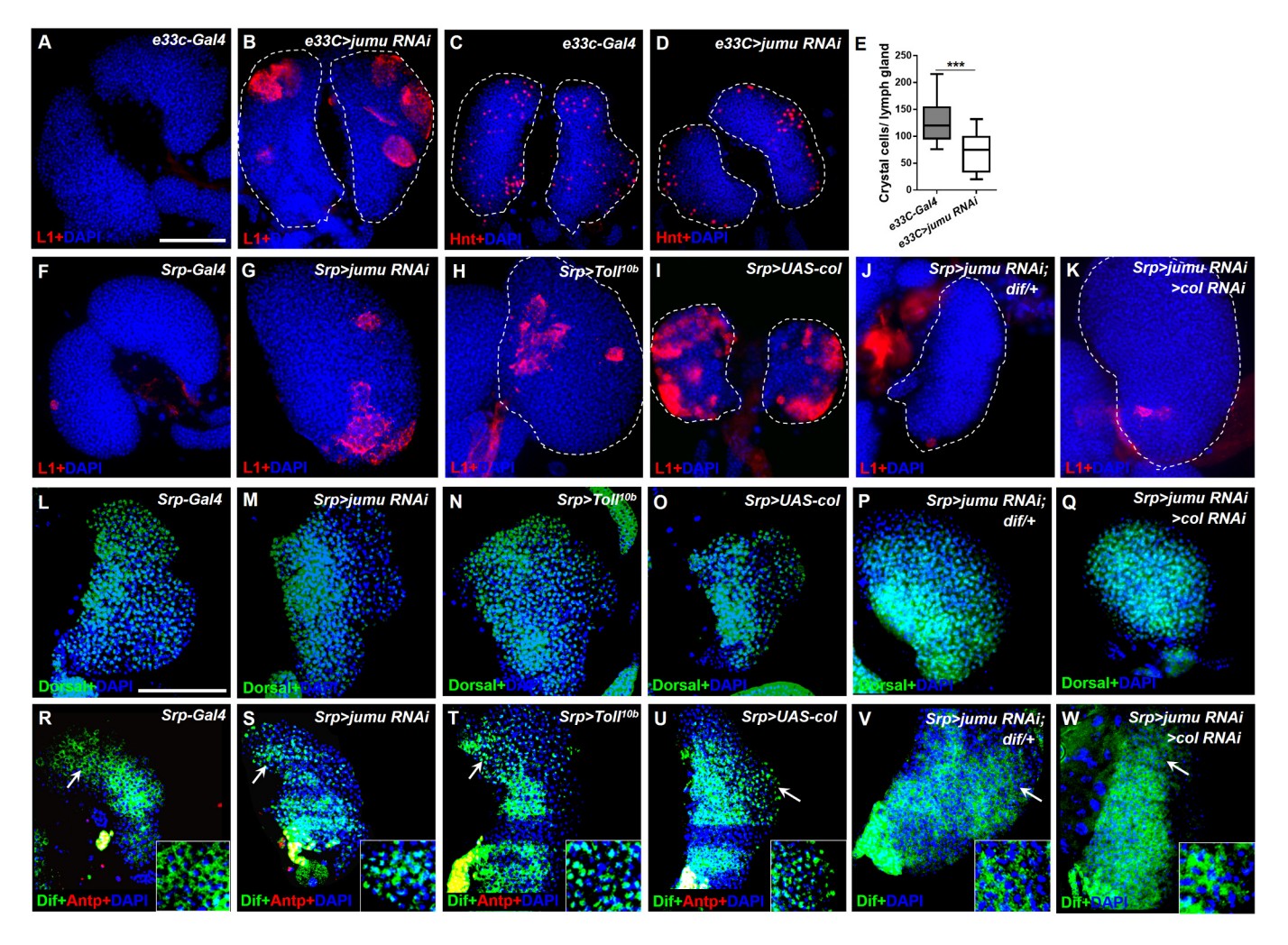

**Figure 9.** Loss of *jumu* in the entire lymph gland induces the generation of lamellocytes through the activation of Toll signaling. (A–D) Immunostaining for L1 (red) and Hnt (red) shows that the knock down of *jumu* in the entire lymph gland increases the number of lamellocytes (B) and reduces the number of crystal cells (D). (E) Quantification of the number of PSC cells (Antp[+] cells). The data are presented in a box-and-whisker representation. ***p<0.001 (Student's *t* test). (F–K) Immunostaining for L1 (red) shows that only less than 20% of lymph glands of the *Srp-Gal4* control display a few lamellocytes (n = 24) (F), but numerous lamellocytes are generated in more than 60% of the lymph glands of *Srp>jumu RNAi* (n = 36) (G) and in more than 80% of the lymph glands of *Srp>Toll^{10b}* (n = 30) (H) and *Srp>UAS-col* (n = 26) (I). The deficiency of *dif* and *col* in *Srp>jumu RNAi* clearly reduces the number of lamellocytes, and only less than 10% (n = 36) (J) and 20% of the lymph glands (n = 32) (K) generated a few lamellocytes, respectively. (L–Q) Immunostaining for Dorsal (green) shows that Dorsal is enriched in nuclei throughout the lymph glands of the control animals (L), and the expression and localization of Dorsal is not changed in the crosses (M–Q). (R–W) Immunostaining for Dif (green) shows that Dif is expressed in the cytoplasm of cells throughout the lymph glands of the control animals (R) but shows nuclear enrichment mainly in the CZ and partly in the MZ of *Srp>jumu RNAi* (S), *Srp>Toll^{10b}* (T) and *Srp>UAS-col* (U). The deficiency of *dif* (V) and *col* (W) in *Srp>jumu RNAi* clearly reduces the nuclear enrichment of Dif. The regions indicated by the arrows are magnified. Dashed white lines outline the edges of the primary lobes. Scale bars: 100 μm.

The following figure supplements are available for figure 9:

**Figure supplement 1.** *jumu* knockdown or *col* overexpression in different specific zones of the lymph gland does not affect lamellocyte differentiation.

**Figure supplement 2.** Analysis of subcellular localization of Dorsal and Dif in circulating cells and the lymph gland.

Dorsal was enriched in nuclei throughout the lymph glands in the control animals (*Srp-Gal4*), whereas Dif was located in the cytoplasm and showed high expression levels in the PSC and lower signals in the MZ and CZ (*Figure 9L,R*). To rule out the effect of the transgenes background of the *Srp-Gal4* line on the localization of Dorsal and Dif, we also detected the expression of Dorsal and Dif in the wild-type (*w1118*) lymph gland and observed the same results (data not shown). The knockdown of *jumu* in the entire lymph gland did not affect the expression and localization of Dorsal but caused an activation of Dif, mainly in the CZ and partly in the MZ but not in the PSC; in fact, most cells in the CZ showed obvious nuclei enrichment of Dif (*Figure 9M,S*; *Figure 9—figure supplement 2E–F'*). In addition, Dorsal and Dif were not enriched in the nuclei of circulating cells of the *Srp>jumu RNAi* mutant (*Figure 9—figure supplement 2A–D*), suggesting that the knockdown of *jumu* might activate the Toll signaling specifically in the lymph gland. We found that the activation of the Toll pathway throughout the through the overexpression of *Toll10b* under the control of *Srp-Gal4* also caused a robust increase in the lamellocyte in more than 80% of the lymph glands (n = 30), and also led to enrichment of Dif in the nuclear in the MZ and CZ without affecting the Dorsal expression and localization (*Figure 9N,T*; *Figure 9—figure supplement 2G,G'*). To further confirm whether Toll signaling is involved in the generation of lamellocytes in the *Srp>jumu RNAi* mutant, we performed rescue experiment with a *dif* mutant. The loss of *dif* effectively reduced the lamellocytes in the *Srp>jumu RNAi* mutant; specifically, only less than 10% of the lymph gland generated a few lamellocytes, and the nuclear enrichment of Dif was abolished (n = 36) (*Figure 9J,V*). Taken together, these results suggest that the activation of Dif in the lymph gland can spontaneously induce lamellocyte differentiation and that Jumu can prevent the activation of Dif and consequently maintain the normal hemocyte differentiation in the lymph gland.

Similar to the phenotype caused by the loss of *jumu* in the lymph gland, a previous study indicated that the ectopic expression of *col* in the entire lymph gland also spontaneously triggers lamellocyte differentiation at the expense of crystal cells spontaneously (*Figure 9I*) (*Crozatier et al., 2004*). Moreover, we found that the forced expression of *col* in different zones of the lymph gland also did not cause an obvious increase in the lamellocytes, with less than 10% of the lymph gland showing a few lamellocytes (n > 30) (*Figure 9—figure supplement 1D–F*). The above results suggest that Jumu negatively regulates Col expression; thus, we asked whether Col is also involved in the activation of Toll signaling. As expected, *col* overexpression in the entire lymph gland also resulted in Dif enrichment, mainly in the nuclei of CZ cells, but did not affect Dorsal expression (*Figure 9O,U*; *Figure 9—figure supplement 2H,H'*). Moreover, the knockdown of *col* in *Srp>jumu RNAi* mutants also inhibited lamellocyte differentiation and nuclear Dif enrichment (*Figure 9K,Q and W*). Altogether, these results demonstrate that the loss of *jumu* in the entire lymph gland might indirectly activate Toll signaling by up-regulating the expression of Col, consequently inducing lamellocyte differentiation.

## Discussion

In recent years, the *Drosophila* lymph gland has been extensively studied as a model for investigating the genetic control of hematopoiesis in flies and humans. The more recent results suggest that the development and maintenance of *Drosophila* hematopoietic progenitors are associated with niche interactions and systemic interactions (*Krzemień et al., 2007*; *Mandal et al., 2007*; *Mercier et al., 2012*; *Shim et al., 2013*). Moreover, some conserved molecules and regulatory networks controlling lymph gland homeostasis have been gradually elucidated.

In this study, we showed that Jumu is expressed in the entire lymph gland and regulates the differentiation of the lymph gland via multiple regulatory mechanisms. It has been demonstrated that Jumu regulates nucleolar morphology and function and chromatin organization and that the correct genetic dose is required for nucleolar integrity and correct nucleolar function (*Hofmann et al., 2010*). Here, we found that Jumu also regulates the hemocyte differentiation of the lymph gland in a dose-dependent manner. The change in the level of *jumu* expression in the entire lymph gland substantially alters the composition of the hemocyte population and the morphology of the lymph gland. The loss of one copy of *jumu* leads to the expansion of Col+ PSC cells and to a reduced differentiation of plasmatocytes and crystal cells (*Figure 10A*). The loss of two copies of *jumu* induces a more severe phenotype of ectopic Col+ PSC cells throughout the primary and secondary lobes of the lymph gland, favors lamellocyte differentiation at the expense of crystal cells, and causes obvious

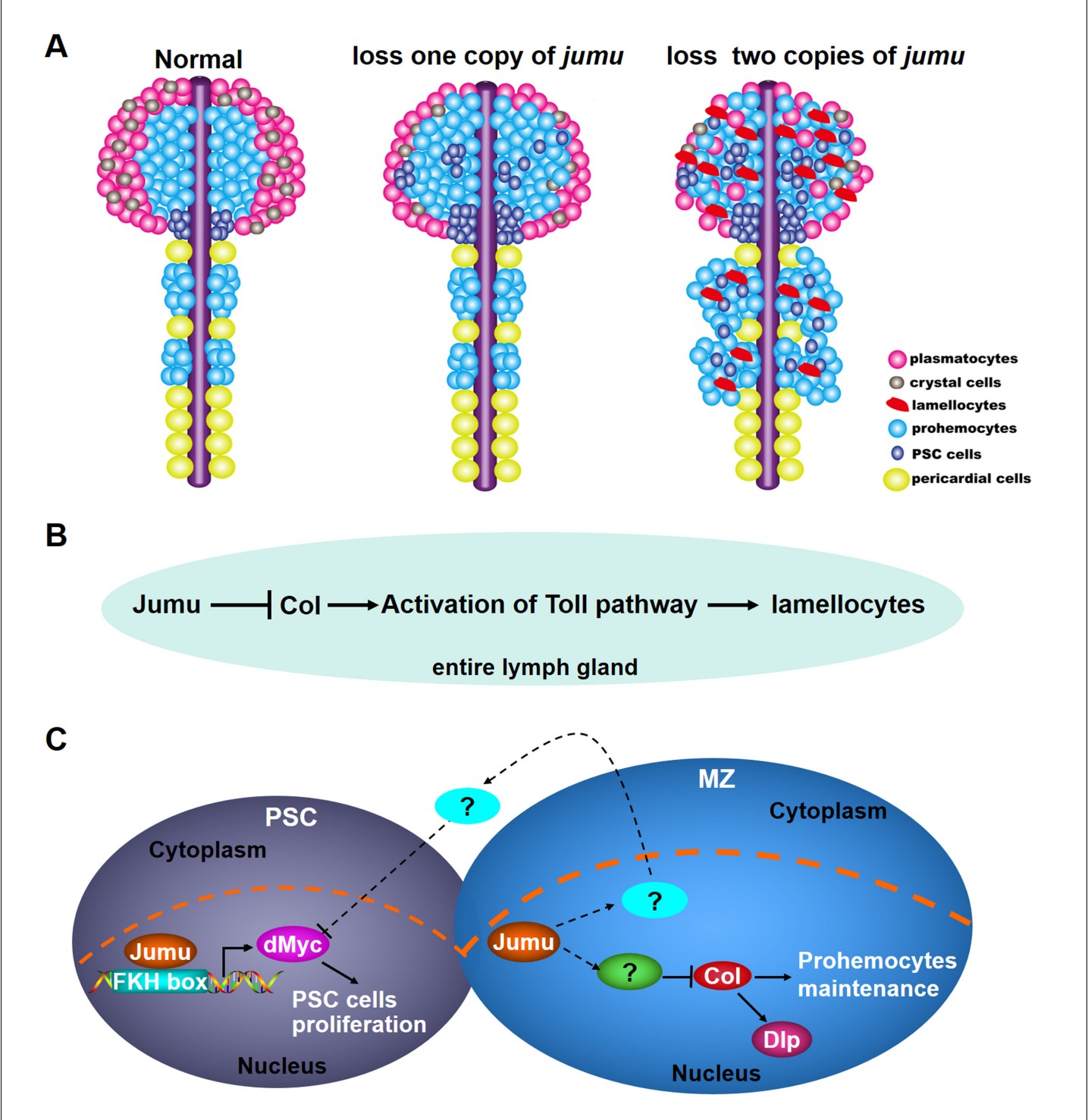

**Figure 10.** Model of the role of Jumu in lymph gland hematopoiesis. (**A**) Schematic representation showing that the fate of hemocyte differentiation is determined by the level of *jumu* expression in the entire lymph gland. The loss of one copy leads to the overproliferation of lymph gland hemocytes, the expansion of PSC cells and a reduction in the differentiation of plasmatocytes and crystal cells. The loss of two copies induces a more severe phenotype of PSC cell expansion throughout the primary and secondary lobes of the lymph gland and favors lamellocyte differentiation. Furthermore, the loss of two copies also leads to the overgrowth of the secondary lobes of the lymph gland. (**B**) Jumu protein in the entire lymph gland prevents the activation of Toll signaling by down-regulating the expression of Col, consequently preventing the differentiation of lamellocytes. (**C**) Schematic representation showing the mechanisms through which Jumu regulates PSC homeostasis and the maintenance of prohemocytes. In the PSC, Jumu controls the proper PSC cell proliferation by regulating the expression of dMyc as a positive transcription factor. In the MZ, Jumu cell-autonomously

*Figure 10 continued on next page*

Figure 10 continued

maintains the differentiation of prohemocytes via the repression of Col levels. Moreover, Jumu *protein* in the MZ prevents the overexpression of dMyc in the PSC in a non-cell-autonomous manner and consequently suppresses the expansion of PSC cells.

overgrowth of the secondary lobes (*Figure 10A*). Notably, the obvious lamellocyte differentiation phenotype was only observed in larvae in which *jumu* was knocked in the entire lymph gland and not in the larvae in which *jumu* was knocked down in different specific zones. This phenomenon suggests that the generation of lamellocytes might require a systemic deficiency of *jumu* in the entire lymph gland and that local *jumu* expression might be sufficient to prevent lamellocyte differentiation. Our results suggest that the loss of *jumu* in the entire lymph gland induces lamellocyte generation through the activation of Toll signaling (*Figure 10B*). Our previous study has showed that the overexpression of *jumu* in both circulating hemocytes and the fat body activates the Toll pathway in hemocytes that were deposited on the fat body, and subsequently leads to the formation of melanotic nodules (*Zhang et al., 2016*). These results suggest that the correct genetic dose of *jumu* is important for the activation of Toll-dependent immune response and chronic inflammation. Moreover, several pathways, such as the JAK/STAT, JNK, Insulin/TOR and Spi/EGFR signaling pathways, have been reported to be directly involved in the differentiation of lamellocytes in different zones of the lymph gland (*Krzemień et al., 2007*; *Owusu-Ansah and Banerjee, 2009*; *Sinenko et al., 2011*). However, although the role of Toll signaling in the immune response and larval hematopoiesis has been investigated widely, most studies focused on the fat body and hemolymph, and few studies investigated the lymph gland. Here, we found that in contrast to its expression in circulating hemocytes, Dorsal is highly expressed in nuclei in the lymph gland, suggesting that it might be involved in lymph gland development and homeostasis in addition to participating in the Toll-dependent immune response. Different from Dorsal, Dif is expressed in the cytoplasm of cells in the lymph gland and becomes enriched in nuclei upon activation of the Toll pathway. This result suggests that similar to its role in the fat body and circulating hemocytes, Dif might also mainly participate in the Toll-mediated immune response. Toll signaling in the PSC also non- cell-autonomously influences fate choice in basal and parasite-activated hematopoiesis (*Gueguen et al., 2013*). In addition, the Toll ligand, Spz protein and *SPE* transcripts are expressed in cells of the lymph gland lobes (*Paddibhatla et al., 2010*; *Mulinari et al., 2006*; *Jang et al., 2006*). These findings indicate that Toll signaling might play an important role in the regulation of lymph gland hematogenesis. However, the regulatory mechanism of Toll signaling on the development and immune response of the lymph gland remain to be addressed. Moreover, we found that ectopic expression of *col* throughout the lymph gland also leads to the activation of Toll signaling and that Jumu might control the Toll signaling by regulating *col* expression (*Figure 10B*). Some elements, such as fly IkB, *cact,* the SUMO conjugase and Ubc9, can balance the activation of immune signaling, the loss of function of these factors result in constitutive activation of Toll signaling in the fat body and circulating hemocytes and indirectly causes chronic inflammation (*Paddibhatla et al., 2010*). Similarly, the correct genetic dose of Jumu and Col might maintain the normal immune homeostasis of the lymph gland and indirectly prevent the development of chronic inflammation. Future studies will address the mechanism through which Jumu and Col regulate the activation of Toll signaling in the lymph gland.

The role of the PSC in prohemocyte maintenance remains controversial. Previous studies suggest that the PSC functions as a hematopoietic niche and is essential for the maintenance of prohemocytes (*Krzemień et al., 2007*; *Mandal et al., 2007*). Conversely, recent studies indicate that ablation of the PSC does not affect prohemocyte maintenance and that the PSC is dispensable for the maintenance of prohemocytes under laboratory breeding conditions (*Benmimoun et al., 2015*; *Oyallon et al., 2016*). However, changes in the levels of certain proteins or signal elements from the PSC indeed affect prohemocyte differentiation. It has been suggested that dMyc is expressed at low levels in PSC cells and that the forced expression of dMyc selectively in the PSC leads to an increased number of PSC cells and further results in an expansion of the prohemocyte pool at the expense of hemocyte differentiation (*Pennetier et al., 2012*). In addition, Slit/Robo signaling, Dpp/Wg signaling and the genetic regulators InR, Bam, and Rbf control the number of PSC cells by regulating the expression of dMyc (*Morin-Poulard et al., 2016*; *Pennetier et al., 2012*; *Sinenko et al., 2009*; *Tokusumi et al., 2015*). Our results indicate that Jumu cell-autonomously ensures proper PSC

cell proliferation by positively regulating dMyc expression in the PSC and that it further affects the maintenance of prohemocytes in an indirect manner (*Figure 10C*). In addition, through an in vitro experiment, we also confirmed that Jumu promotes the transcription of the *dmyc* gene via directly binding to its promoter. Moreover, Jumu shows no obvious regulatory effect on the dMyc level in the MZ and CZ, which suggests that the role of Jumu in the regulation of dMyc expression is zone-specific. In addition to the cell-autonomous mechanism, Jumu also controls the numbers of PSC cells in a non-cell-autonomous manner. The loss of *jumu* in the entire lymph gland or the MZ can lead to the ectopic expansion of PSC cells, and this effect is accompanied by the overexpression of dMyc in these cells. This result suggests that Jumu might regulate the expression of some regulatory factors in the MZ that can deliver signals to neighboring PSC cells and indirectly prevent the overexpression of dMyc, thereby maintaining the proper PSC cell number and location (*Figure 10C*). However, over-expression of *jumu* in the MZ alone is not sufficient to reduce dMyc expression and the number of PSC cells. In fact, this result indicates that Jumu protein in the MZ might mainly function in the main-tenance of the normal development margin between the MZ and the PSC and further maintains the homeostasis of each zone of the lymph gland but does not directly regulate proliferation in the PSC in a spontaneous manner. It has been suggested that Slit from the cardiac tube controls PSC mor-phology and that inter-organ crosstalk between the cardiac tube and the PSC is required to preserve the normal PSC morphology (*Morin-Poulard et al., 2016*). Similarly, Jumu might also participate in the inter-zone communication between the MZ and the PSC and thereby preserve the normal devel-opment of each zone. Notably, different from the PSC phenotypes caused by the loss of *jumu* in the MZ or in the entire lymph gland, the additional PSC cells induced by the overexpression of *jumu* or *dmyc* in the PSC are all located in the original PSC domain rather than being scattered through the MZ or CZ, further strengthening the notion that Jumu protein in the MZ and the PSC regulate the distribution of PSC cells through different mechanisms. The relationship between Jumu and the known signaling pathways associated with PSC development remains to be addressed.

In addition to regulating the homeostasis of the PSC, Jumu also maintains prohemocyte differen-tiation in a cell-autonomous manner by preventing the overexpression of Col in the MZ (*Figure 10C*). More recently, it was shown that the EBF transcription factor Col is also expressed at low levels in prohemocytes as well as in the PSC and that Col directly promotes the maintenance of *Drosophila* blood cell progenitors independently of the niche (*Benmimoun et al., 2015*; *Oyallon et al., 2016*). The overexpression of *jumu* in the MZ is sufficient to reduce the Col level and induce an increased differentiation of prohemocytes. However, we found that although the knock-down of *jumu* in the MZ causes an increase in the Col level, the differentiation of plasmatocytes and crystal cells is not markedly reduced. We speculate that this phenomenon might be caused by the dose-dependent function of *jumu* and that the RNAi-induced decrease in the *jumu* expression level in the MZ is insufficient to obtain a sufficient expression of Col to inhibit the differentiation of prohe-mocytes. Moreover, it has been suggested that Col protein in the PSC controls the number of PSC cells by activating the expression of Dally-like (Dlp) (*Pennetier et al., 2012*). In this study, we showed that Col also positively regulates the expression of Dlp in the MZ (*Figure 10C*). Different from its role in the MZ, ectopic expression of *col* throughout the lymph gland leads to the generation of lamellocytes and reduced crystal cells but does not affect plasmatocyte differentiation (*Crozatier et al., 2004*); interestingly, these phenotypes were also observed in *jumu* double hetero-zygotes but not in *jumu* heterozygotes. These results suggest that the correct expression levels and regions of *jumu* or *col* are required for the normal lymph gland development. Further work is needed to decipher the regulation and the mechanisms of action of Col in the maintenance of lymph gland homeostasis. Moreover, the signals associated with *jumu* and other transcription factors that regulate *col* expression remain to be addressed. A recent study showed that Ebf2, the mammalian ortholog of Col, can promote hematopoietic stem cell (HSC) maintenance by controlling the devel-opment of osteoblastic cells (*Kieslinger et al., 2010*). Moreover, it has been shown that the mem-bers of the forkhead-box (FOX) transcription factor family in human and mouse play crucial roles in various aspects of immune regulation, from lymphocyte survival to thymic and embryonic develop-ment and cell cycle regulation (*Coffer and Burgering, 2004*; *Lehmann et al., 2003*; *von Both et al., 2004*; *Martínez-Gac et al., 2004*). Foxn1, the mammalian ortholog of Jumu, has been shown to reg-ulate the growth and differentiation of thymic epithelial cells in mouse. *Foxn1*-knockout mice show congenital hairlessness, severe immunodeficiency and a rudimentary thymus with a lack of T-cell development (*Nehls et al., 1994*; *Balciunaite et al., 2002*; *Su et al., 2003*). Therefore, our present

results provide a better understanding of the regulation of *Drosophila* hematopoiesis and important insights into the regulatory mechanisms of the EBF and FOX families of transcription factors in the mammalian hematopoietic niche.

## Materials and methods

### *Drosophila* strains

The following mutant alleles, deficiencies, and transgenes were used: $jumu^{GE27806}$ and $shg^{GE13814}$ were purchased from GenExel (Daejeon, South Korea); $jumu^{Df2.12}$, $jumu^{Df3.4}$ and *UAS-jumu* were gifts from Alan M. Michelson (**Ahmad et al., 2012**; **Zhu et al., 2012**); *Hml-Gal4 UAS-2xEGFP*, *dome-less-Gal4 UAS-2xEGFP*, $col^1$/*CyO;P(col5-cDNA)/TM6B, Tb, Antp-Gal4/TM3, Sb* and *Srp-Gal4* were gifts from Utpal Banerjee (**Mandal et al., 2007**); *col-Gal4 UAS-mCD8GFP*, *domeMESO*, *hhF4-GFP*, *UAS-col* and *UAS-col RNAi* were gifts from Lucas Waltzera (**Benmimoun et al., 2015**); *P{en2.4-GAL4}e33C* (*e33C-Gal4*) was a gift from Dominique Ferrandon; *Tep-IV-Gal4* was a gift from the Kyoto Stock Center (DGRC); *UAS-jumu RNAi* ($jumu^{GD4099}$) was obtained from the Vienna *Drosophila* RNAi Center (VDRC); *Tub-Gal4, Tub-Gal80$^{ts}$* (*Tub-Gal4$^{ts}$*), *UAS-dmyc RNAi* (JF01761) and *elav-Gal4* were obtained from the Tsinghua Fly Center; *Hand-Gal4* was a gift from Wuzhou Yuan; *cdc42-DN* was obtained from Bloomington; *HHLT-Gal4* was a gift from Jiwon Shim; and *UAS-Toll$^{10b}$* and *dif* were gifts from Bruno Lemaitre. The $w^{1118}$ line or the respective UAS or Gal4 parent lines were used as controls when required. The crosses involving RNAi lines or *Tub-Gal4$^{ts}$* were reared at 29°C, and the other strains and crosses were reared at 25°C. All strains and crosses were cultured on standard cornmeal-yeast medium under a 12 hr light/12 hr dark cycle.

### Antibody production

A fusion protein containing a nonconserved portion of the deduced Jumu protein (residues 141–330) was produced in *E. coli* using the pRSETA (6x-His) vector from Qiagen. The fusion protein was purified using Ni-NTA resin and used to immunize rats.

### Immunohistochemistry and ROS detection

For antibody staining, the lymph glands were dissected in PBS, fixed with 3.7% formaldehyde in PBS for 20 min, pre-incubated in blocking solution (PBS with 0.1% Tween-20% and 5% goat serum) and then incubated with primary antiserum diluted in blocking solution. The following primary antibodies were used: mouse anti-P1, mouse-anti-L1 (gifts from I. Ando), mouse anti-Hnt (AB_528278), mouse anti-Ptc (AB_528441), Rat anti-Shg (AB_528120), mouse anti-Antp (AB_528082), mouse anti-Dlp (AB_528191) (Developmental Studies Hybridoma Bank), mouse anti-Col (gift from M. Crozatier) (**Pennetier et al., 2012**), mouse anti-$\beta$-galactosidase (Sigma), rat anti-Jumu, rabbit anti-dMyc (Santa Cruz), and mouse anti-P-Mad (gift from Ed Laufer, USA). Alexa Fluor 488-, Alexa Fluor 568- and Alexa Fluor 594-conjugated secondary antibodies (Molecular Probes) were used. ROS detection was performed as previously described (**Evans et al., 2014**) using dihydroethidium (Invitrogen). Images were obtained using a Zeiss LSM510 confocal microscope or a Zeiss Axioplan 2 microscope equipped with fluorescence optics. All staining was performed in samples from at least three independent experiments.

### Image analyses

For quantifications, the lymph glands were scanned at an optimized number of slices (aiming to acquire all staining signals throughout the tissue) using a Zeiss Axioplan 2 microscope, usually less than three optical sections from the Z stack can contain all signal from one tissue. To quantify the crystal cell and ROS indexes and the PSC cell number, the consecutive slices from the same primary lobe were assembled in Photoshop CS6, and the merged images were used for the quantifications. The crystal cell and ROS indexes were calculated as the total number of Hnt$^+$ or ROS$^+$ cells (quantified using ImageJ, NIH), respectively, in each primary lobe divided by the relative area of the primary lobe. The total PSC cell (Antp$^+$ cell) number in each primary lobe was also quantified using ImageJ. At least 30 lymph glands were scored for each genotype. For the quantification of P1$^+$ cells and the Shg$^+$ and domeMESO$^+$ cell indexes, the fluorescent images collected by Zeiss Axioplan 2 microscope were converted to 8-bit images and then recalibrated to obtain an identical threshold using

ImageJ. The area with the identical threshold was captured and measured with ImageJ. To measure the total size of the primary lobes, the DAPI-expressing area was captured using freehand selection tool of ImageJ and measured. The plasmatocyte differentiation index was assessed by determining the average area of P1$^+$ cells from all optical sections of a primary lobe compared with the area of the primary lobe. The proportion of prohemocytes in the primary lobes (MZ/lobe 1) was defined as the average area of Shg$^+$ or domeMESO$^+$ cells in all optical sections of a primary lobe compared with the area of the primary lobe. At least 30 lymph glands were analyzed for each genotype. For quantification of the Col, Dlp, dMyc and Jumu fluorescence signal intensities in the MZ or PSC, images were captured with a Zeiss LSM510 confocal microscope at the same exposure times, and at least three confocal slides (4 µm) per stack were analyzed. The total intensity values for Col, Dlp, dMyc and Jumu in each ROI (region of interest) were measured using ImageJ. For the analysis of Col and Dlp staining in the MZ, dome-GFP was used to define the ROIs. For the analysis of dMyc and Jumu staining in the PSC, Antp labeling was used for to define the ROIs. The ROIs in the fluorescent images were captured using the freehand tool and then converted to 8-bit images. The total intensity value of the ROIs with an identical threshold was captured and measured with ImageJ. The average pixel intensity values of dMyc or Jumu in PSC cells (integrated intensity/area of Antp-labeled cells) were calculated. The pixel intensity values of Col or Dlp in the MZ were assessed as the integrated intensity in the ROIs compared with the area of the primary lobe. For each fluorescence quantification experiment, at least 20 lymph glands were analyzed.

## Chromatin immunoprecipitation (ChIP) assays

ChIP assays were performed as described previously (*Kim et al., 2005*). In brief, S2 cells (CVCL_Z232, Invitrogen, Cat#R690-07) were transfected with the pMK33-Flag-jumu full CDS construct using the Effectene Transfection kit (Qiagen). S2 cells stably expressing Flag-jumu were fixed in 1% formaldehyde at room temperature. The samples were then sonicated to obtain DNA fragments between 200 and 800 bp in length. The fragmented chromatin was incubated with the anti-Flag antibody (Sigma) or IgG (Santa Cruz) bound to Salmon Sperm DNA/Protein A Agarose Slurry (Upstate) overnight at 4°C. The immunoprecipitated DNA fragments were recovered using a MiniElute purification kit (Qiagen) and analyzed by PCR amplification. Similar results were obtained in three independent ChIP experiments. The primer sequences are shown in *Supplementary file 1*.

## Construction of *dmyc* promoter reporter plasmid and site-directed mutagenesis

The *dmyc* promoter −643 to +467 was cloned into the pGL3 basic vector (firefly luciferase reporter plasmid, Promega) using *KpnI* and *XhoI* sites. The two base mutations of FKH-binding sites in the *dmyc* promoter were generated by the overlapping PCR method (*Vallejo et al., 2008*). The recombinant pGL3 plasmid with the *dmyc* promoter −643 to +467 was used as a template. The first and second rounds of PCR were performed using a primer pair composed of the forward primer amplifying *dmyc* promoter −643 to +467 (F) and the reverse mutated primer containing the two base exchanges in the FKH-binding sites (M1R, M2R or M3R) of the *dmyc* promoter and a second primer pair corresponding to the complementary forward mutated primer (M1F, M2F, or M3F) and the reverse primer amplifying *dmyc* promoter −643 to +467 (R), respectively. The two amplified overlapping fragments were used as templates for a subsequent overlap extension PCR with the primer pair amplifying the *dmyc* promoter −643 to +467 (F+R). The primer sequences used are shown in *Supplementary file 1*. The PCR fragment containing the mutated FKH-binding site was then cloned into the pGL3 basic vector. The introduction of the mutations was verified by DNA sequencing.

## Transfection and luciferase reporter gene assay

A total of $5 \times 10^5$ S2 cells (CVCL_Z232, Invitrogen, Cat#R690-07) stably expressing the pMK33-Flag-jumu full CDS construct or empty pMK33 vector were plated per well in a 24-well plate and then co-transfected with 200 ng of *dmyc* promoter firefly *luciferase* reporter plasmid and 20 ng of Renilla *luciferase* pRL-TK reporter vector. The cells were harvested after 48 hr of incubation. Luciferase assays were performed using the Dual-Luciferase Reporter Assay System (Promega), and the normalized luciferase values were calculated as the ratio of the firefly to the Renilla luciferase activities.

Plasmid transfection was performed in at least triplicate, and luciferase activity was measured in duplicate or triplicate.

## Quantitative real-time PCR

The total RNA from the entire third-instar larvae or the dissected hemocytes (from 300 to 500 third-instar larvae) was isolated with TRIzol (Invitrogen). The obtained total RNA was used to generate cDNA with M-MLV Reverse Transcriptase (Promega). The real-time PCR amplification was performed using SYBR® Select Master Mix (Applied Biosystems) on a Roche 4500 real-time PCR system. The results were normalized to the level of *RpL32* mRNA in each sample. Three experiments per genotype were averaged. A biological replicate was performed, and the same results were obtained. The primer sequences used are shown in *Supplementary file 1*.

## Statistical analysis

The statistical analyses were performed through two-tailed unpaired Student's t-tests or one-way ANOVAs using GraphPad Prism software. The threshold for statistical significance was established as *$p<0.1$, **$p<0.01$ and ***$p<0.001$.

## Acknowledgements

We thank Alan M Michelson, Utpal Banerjee, Lucas Waltzera, Dominique Ferrandon, Jiwon Shim, Wuzhou Yuan and Bruno Lemaitre for providing the numerous fly strains used in this study. We acknowledge István Andó, Michèle Crozatier, and Ed Laufer for providing the anti-P1, anti-L1, anti-Col, and anti-P-Mad antibodies. We thank the GenExel *Drosophila* Stock Center, Kyoto *Drosophila* Genetic Resource Center, Vienna *Drosophila* RNAi Center, and Tsinghua Fly Center for sharing the numerous fly stocks utilized in this research. We also thank Michèle Crozatier for the valuable suggestions provided regarding this work. This work was supported by the National Natural Science Foundation of China (31270923).

## Additional information

### Funding

| Funder | Grant reference number | Author |
|---|---|---|
| National Natural Science Foundation of China | General Program 31270923 | Li Hua Jin |

The funders had no role in study design, data collection and interpretation, or the decision to submit the work for publication.

### Author contributions

YH, Investigation, Visualization, Writing—original draft; LHJ, Supervision, Funding acquisition, Project administration, Writing—review and editing

### Author ORCIDs

Li Hua Jin, http://orcid.org/0000-0001-5912-9800

## Additional files

### Supplementary files

• Supplementary file 1. PCR primer sequences.

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
