## [Decision Letter]

[Editors’ note: a previous version of this study was rejected after peer review, but the authors submitted for reconsideration. The first decision letter after peer review is shown below.]

Thank you for submitting your work entitled "Dual role for Jumu in the control of hematopoietic progenitors in the *Drosophila* lymph gland" for consideration by *eLife*. Your article has been favorably evaluated by K VijayRaghavan (Senior Editor) and three reviewers, one of whom is a member of our Board of Reviewing Editors. The following individual involved in review of your submission has agreed to reveal their identity: Jiwon Shim (Reviewer #2).

Our decision has been reached after consultation between the reviewers. Based on these discussions and the individual reviews below, we regret to inform you that your work will not be considered further for publication in *eLife*.

The exact verbatim reviewer's comments are attached below. As you will see, the reviewer's initial reaction to the paper is upbeat and they suggest many individual experiments that we hope will be useful to you to enhance the manuscript. But in the follow up discussion, it became clear that there are too many questions raised by the reviewers, and particularly given that much of the criticism is over whether a proper mechanistic basis has been established, we do not feel that a revision within the *eLife* timing and standards can be reached. You do have the option of resubmission, but that will be evaluated as a fresh manuscript on the basis of relevance, advancement and in light of the reviews for this version.

*Reviewer #1:*

I was excited about this paper when it was first sent to me. It was particularly intriguing to read about the autonomous and non-autonomous signals that the authors mention in the Abstract. Upon further scrutiny, I am not as enthusiastic as I was before in terms of depth and novelty of this work and although I can be convinced otherwise by other expert reviewers, I do not see this manuscript moving the field much farther.

Jumu/Domina was identified as a position effect variegation causing mutation a couple of decades ago. The authors are the first to investigate its role in hematopoiesis. In their previous paper, the emphasis was on producing melanotic tumors and through unknown mechanisms activating Toll signaling. These results are not properly described here. Activation of Toll should have a big effect on the lymph gland as well. Perhaps the extra lamellocytes in the homozygote are related to this effect?

Given what is known about this gene, it is expected to participate in the regulation of large numbers of genes and the zone-specific phenotypes coupled with ubiquitous expression pattern would suggest that the developmental context is provided by signals, partners, regulators and epigenetic status and not by the Jumu protein itself. Understanding the hierarchies and interactions between the various pathway members generates mechanisms. Unfortunately, other than myc regulation (already shown by Crozatier's laboratory to control PSC size; Mourin-Pollard et al., Nature communication), the paper is to be viewed as a very preliminary phenotypic analysis mostly without mechanisms. The is not even an attempt to identify the non-autonomous MZ to PSC signal which is what got me really interested after reading the abstract. As shown, the authors imply a morphogen involved in the process. A handful has been described and could easily have been tested with available tools.

1) The data suggest Jumu functions downstream of Antp but upstream of Col and Hh. Does this allow some ordering of events? When does Jumu expression begin? Is it first expressed in the PSC, later expanding to the rest of the gland? Richard Mann had shown fkh is controlled by the Hox gene Scr and Exd. In other contexts, Antp can do that as well. An interesting interplay between Antp and Homothorax has been reported in the lymph gland. Is that related to Jumu expression?

2) Why were only the heterozygotes used in Figure 1? It is understandable that homozygous null is lethal, but the transheterozygote has to show these phenotypes as well.

3) Why does ROS go down as the MZ expands?

4) I have a great deal of difficulty understanding the quantification data throughout the paper. It seems that what is being quantitated is different in each figure such as number of cells, pixel density or staining index. There is no clear mention of how much of the tissue is quantitated. The methods describe that an optimal number of sections were used. What does this mean? Was the number of optical sections kept constant in each case? Indeed, are these all confocal images? An LSM510 is mentioned but most of the analysis was done on an Axioplan2 microscope. If the entire depth of the tissue is imaged, there will be a great deal of variability. How does pixel intensity converted to number of cells when the intensity per cell could vary and how were the thresholds set? Why are the quantitations in the later figures represented as histograms? Ideally, the entire paper should have a uniform standard in how images are quantified so that the readers can get a clear comparison between all of the data presented. Perhaps that is what was done, but it does not come through that way.

5) In Figure 2., the data look very nice, but a developmental series will be useful in knowing whether the cells are migrating or arising in the wrong site. Overexpression of Antp causes an expansion of the PSC, the reason has never been clear. This again brings up the question of whether ANTP controls JUMU.

6) Figure 3. It would be good to see the genotype in panel B stained for MZ and CZ markers.

7) Figure 8., how does panel 1 match with the results in the authors' previous paper?

8) Also Figure 8, the binding assay needs a critical control with mutated binding sites.

*Reviewer #2:*

Jin et al. describes dual functions of Jumu in the regulation of blood progenitors in both the MZ and in the PSC. Authors have shown that one copy loss of Jumu inhibits the blood progenitor differentiation due to the expansion of PSC cells. These phenotypes are restored by expression of Jumu specifically in the blood progenitors, while the other cell-type specific drivers do not result in the rescue phenotype. Authors have also shown that overexpression of Jumu in the blood progenitors induces differentiation of mature blood cells, and confirmed that expression of Col and Dlp are regulated by levels of Jumu in the MZ. In addition to the function of Jumu in the MZ, authors have identified that Jumu in the PSC controls the proliferation of PSC cells and thereby indirectly affects the progenitor maintenance. One possible downstream of Jumu identified in this paper is dMyc, of which expression can be directly controlled by Jumu. Overall, these data suggest two independent functions of Jumu in the regulation of progenitors by inhibition of Col in the MZ or by activating dMyc expression in the PSC.

1) One of the main findings of Jin and Hao is that one-copy loss of Jumu induces expansion of the PSC cells in the lymph gland. However, the phenotype shown by Jumu heterozygote (Figure 2' or H) includes both proliferation and clustering defects of the PSC. Previous study has shown that these two phenomena are regulated by two independent pathways in the PSC; proliferation is regulated by Dpp/dMyc and clustering, by DE-cad/Cdc42 (Morin-Poulard et al., 2015). Authors have demonstrated that expression of Jumu in the MZ rescues both proliferation and clustering phenotypes of the PSC. To support the authors' claim, it will be important to identify genetic correlations in between Jumu in the MZ and two other pathways in the PSC in detail.

2) Along the same lines; if Jumu is widely expressed in other tissues including the dorsal vessel, it is possible that Jumu regulates Slit/Robo that in turn regulates both proliferation and clustering of the PSC. To exclude the involvement of Slit/Robo pathway and its indirect regulation, experiments such as Hand-gal4 driving Jumu RNAi or overexpression of Jumu will simply solve this concern.

3) One-copy loss of Jumu induces expansion of PSC cells in the lymph gland (Figure 2), and this phenotype is recapitulated by Dome-gal4 driving Jumu RNAi (Figure 4). However, downregulation of Jumu in the PSC further reduces the number of PSC cells, while overexpression of Jumu leads to the proliferation of PSC cells. These data indicate that Jumu regulates dMyc in opposite ways in two zones; Jumu in the MZ non-autonomously inhibits dMyc whereas Jumu in the PSC autonomously activates dMyc. However, in the development of the PSC, PSC cells require proliferation at earlier stages while Jumu needs to inhibit the proliferation in later instars. Developmental relevance of Jumu would be also important for understanding the gene function. Supplementing strong evidence that there are changes in the expression levels of Jumu/dMyc throughout larval development, or different phenotypes led by early vs late knockdown of Jumu in different zones would strengthen the developmental function of Jumu/dMyc in the PSC.

*Reviewer #3:*

In this study, Hao and Lin explore the role of Jumu in controlling blood progenitor function in *Drosophila*, specifically in the larval lymph gland. Their findings demonstrate that Jumu maintains prohemocytes by cell-autonomously preventing the overexpression of Col in the MZ and by non-cell-autonomously inhibiting dMyc in the PSC to regulate its size. Their study also studies reveals a requirement for Jumu in cell-autonomously controlling PSC cell proliferation via dMyc. Their findings provide novel insight into the molecular regulation of progenitor maintenance, in particular a novel mechanism in which the progenitor population provides feedback to the cells of the niche/PSC for their own maintenance. The quality of the data is excellent and consistent with established standards for this field. However, there are a few concerns that need to be addressed:

Expression of Hedgehog in the PSC should be assessed upon overexpression of Jumu in the MZ to determine if the increased differentiation observed is due to loss of this maintenance signal.

It cannot be assumed that lamellocytes present in the lymph gland of trans-heterozygote Jumu mutants is due to loss of collier (Discussion, second paragraph). Staining of lymph glands for lamellocytes in which Jumu is removed in specific zones would allow insight into the process.

Dome-gal4 is expressed outside of the MZ in many other organs and the authors cannot rule an effect external to the signal responsible for the observed effect of inhibiting PSC expansion in Jumu mutants. This should be stated.

It would be useful to have clones of Jumu made in which the cell autonomous effects observed can be established, particularly given that the levels of Jumu may have different effects on cell maintenance and differentiation.

In addition, the following need to be corrected prior to publication:

Figure 1: the ROS staining looks poorly done.

Figure 2: these experiments need labeling with a MZ marker like shg or ptc.

Figure 2: it is not clear if the ectopic Antp cells are derived from the PSC or the result of a homeotic transformation in which an ectopic PSC is generated. The fact that Dome>UAS-Jumu rescues the phenotype of ectopic PSC may have to do with Jumu preventing the PSC fate in the MZ (domeless is expressed very early in lymph gland development) rather than the PSC increasing in size.

Figure 2: The quantification panel can go in supplementary to maintain symmetry for the figure.

Figure 3: the Jumu staining looks cytoplasmic while clearly nuclear with some cytoplasmic in the PSC. Have the authors looked at the developmental pattern of expression of Jumu to determine if it is ever nuclear in the lymph gland outside of the PSC? The developmental pattern of Jumu expression is essential to understand its function.

Figure 3: images too blurry.

Figure 3: can go in supplementary.

Figure 4: the figure is missing dome-gal4, UAS-Jumu, UAS-col with Antp immunostaining.

Figure 9: the schematic model is misleading since the trans heterozygote must include a hypomorphic mutation and not a complete loss of function to allow survival into adulthood. Please modify the description to trans heterozygote.

---

## [Author Response]

[Editors’ note: the author responses to the first round of peer review follow.]

*Reviewer #1:*

*I was excited about this paper when it was first sent to me. It was particularly intriguing to read about the autonomous and non-autonomous signals that the authors mention in the Abstract. Upon further scrutiny, I am not as enthusiastic as I was before in terms of depth and novelty of this work and although I can be convinced otherwise by other expert reviewers, I do not see this manuscript moving the field much farther.*

We thank the reviewer for these constructive comments. We have performed an in-depth analysis of the regulation of PSC morphology by Jumu in the MZ and the mechanism of lamellocyte differentiation in the subsections “The loss of jumu in the MZ non-cell-autonomously induces increases in dMyc in the PSC” and “The loss of jumu in the entire lymph gland leads to the generation of lamellocytes through the activation of Toll signaling”, respectively. We believe that this additional information will improve the interest of this manuscript.

*Jumu/Domina was identified as a position effect variegation causing mutation a couple of decades ago. The authors are the first to investigate its role in hematopoiesis. In their previous paper, the emphasis was on producing melanotic tumors and through unknown mechanisms activating Toll signaling. These results are not properly described here. Activation of Toll should have a big effect on the lymph gland as well. Perhaps the extra lamellocytes in the homozygote are related to this effect?*

Thank you for the advice. Our previous paper indicated that the overexpression of *jumu* simultaneously in both hemocytes and the fat body leads to an enrichment of Dorsal and Dif in the nuclei of aggregated circulating hemocytes and deposited hemocytes on the fat body. However, the Dif/Dorsal levels did not increase in the lymph gland. Here, we used the lymph gland-specific driver *Srp-Gal4* to knock down *jumu* and detected the activation of Toll signaling in the lymph gland using Dorsal and Dif antibodies. Surprisingly, we found that Dorsal is enriched in nuclei throughout the lymph glands in the control, whereas Dif is located in the cytoplasm and showed high expression levels in the PSC and lower signals in the MZ and CZ. Moreover, Dif was found to be activated and mainly enriched in nuclei in the MZ and CZ of the lymph gland in *Srp>jumu RNAi and Srp> Toll^10b^*(Figure 9). In addition, the loss of *dif* effectively reduced the lamellocytes in *Srp>jumu RNAi* (Figure 9). These results suggest that the loss of *jumu* in the lymph gland activates Toll signaling and thereby induces the generation of lamellocytes. We have described these results in the subsection “The loss of jumu in the entire lymph gland leads to the generation of lamellocytes through the activation of Toll signaling”.

*Given what is known about this gene, it is expected to participate in the regulation of large numbers of genes and the zone-specific phenotypes coupled with ubiquitous expression pattern would suggest that the developmental context is provided by signals, partners, regulators and epigenetic status and not by the Jumu protein itself. Understanding the hierarchies and interactions between the various pathway members generates mechanisms. Unfortunately, other than myc regulation (already shown by Crozatier's laboratory to control PSC size; Mourin-Pollard et al., Nature communication), the paper is to be viewed as a very preliminary phenotypic analysis mostly without mechanisms. The is not even an attempt to identify the non-autonomous MZ to PSC signal which is what got me really interested after reading the abstract. As shown, the authors imply a morphogen involved in the process. A handful has been described and could easily have been tested with available tools.*

Thank you for the advice, We have performed an in-depth analysis of the relationship between *jumu* and Slit/Rob signaling in the control of the PSC morphology and found that Jumu in the MZ controls PSC morphology by indirectly regulating dMyc expression in PSC cells through a Slit/Robo and BMP/Dpp-independent mechanism in an non-autonomous manner (Figure 5 and Figure 5—figure supplement 1; subsection “The loss of jumu in the MZ non-cell-autonomously induces increases in dMyc in the PSC”).

*1) The data suggest Jumu functions downstream of Antp but upstream of Col and Hh. Does this allow some ordering of events? When does Jumu expression begin? Is it first expressed in the PSC, later expanding to the rest of the gland? Richard Mann had shown fkh is controlled by the Hox gene Scr and Exd. In other contexts, Antp can do that as well. An interesting interplay between Antp and Homothorax has been reported in the lymph gland. Is that related to Jumu expression?*

Thank you for the advice. We did not mention that *jumu* functions downstream of Antp but rather suggested that *jumu* affects the proliferation and expansion of Col^+^, Antp^+^ and HH^+^ PSC cells and that the loss of *col* can inhibit the phenotype caused by *jumu* deficiency.

With respect to *jumu* expression, previous studies have shown that *jumu* beings to be expressed at the embryo stage. (“Cheah PY, Chia W, Yang X. 2000. Jumeaux, a novel *Drosophila* winged-helix family protein, is required for generating asymmetric sibling neuronalcell fates. Development 127: 3325–3335.” and “Strödicke M, Karberg S, Korge G. 2000. Domina (Dom), a new *Drosophila* member of the FKH/WH gene family, affects morphogenesis and is a suppressor of position-effect variegation. Mech Dev 96: 67–78.”). In addition, we detected the expression pattern of Jumu protein in the lymph gland at different larval stages through staining with anti-Jumu antibody and observed that Jumu is expressed in the entire lymph gland and mainly localized in the nuclei (Figure 4”; subsection “Jumu expression in prohemocytes is required for preventing the ectopic expansion of PSC cells”).

With respect to the known role of Hox gene Homothorax (Hth) in Antp expression is restricted to embryo, Antp^+^ cells were decreased in the embryo of the Homothorax (Hth) mutant (Mandal L, Martinez-Agosto JA, Evans CJ, Hartenstein V, Banerjee U. 2007. A Hedgehog- and Antennapedia-dependent niche maintains *Drosophila* haematopoietic precursors. Nature 446: 320–324.); however, Mandal et al.did not investigate the larvae stage. Therefore, we knocked down *hth* using the PSC-specific driver *col-Gal* and found that Antp^+^ cells presented no difference between the *col>hth RNAi* and control PSCs (the data was shown in Figure 11). These results suggest that different from the function of Jumu, Hth might not directly participate in PSC development during the larval stage.

Author response image 1.**DOI:**
http://dx.doi.org/10.7554/eLife.25094.022

*2) Why were only the heterozygotes used in Figure 1? It is understandable that homozygous null is lethal, but the transheterozygote has to show these phenotypes as well.*

We added the trans-heterozygote detection data to Figure 1 and Figure 2, which reveal a significant phenotype different compared with heterozygotes.

*3) Why does ROS go down as the MZ expands?*

ROS is different from other MZ markers, such as dome, Shg and Ptc, which prevent prohemocyte differentiation. ROS only appear in the MZ until the middle- third larval stage and can primes hematopoietic progenitors for differentiation (Owusu-Ansah E, Banerjee U. 2009. Reactive oxygen species prime *Drosophila* hematopoietic progenitors for differentiation. Nature461: 537–541”). We observed low ROS levels in *jumu* mutant third-instar larva; therefore, the progenitor cells are unable to differentiate into mature hemocytes. This result is also consistent with the reduced CZ detected in the *jumu* mutants.

*4) I have a great deal of difficulty understanding the quantification data throughout the paper. It seems that what is being quantitated is different in each figure such as number of cells, pixel density or staining index. There is no clear mention of how much of the tissue is quantitated. The methods describe that an optimal number of sections were used. What does this mean? Was the number of optical sections kept constant in each case? Indeed, are these all confocal images? An LSM510 is mentioned but most of the analysis was done on an Axioplan2 microscope. If the entire depth of the tissue is imaged, there will be a great deal of variability. How does pixel intensity converted to number of cells when the intensity per cell could vary and how were the thresholds set? Why are the quantitations in the later figures represented as histograms? Ideally, the entire paper should have a uniform standard in how images are quantified so that the readers can get a clear comparison between all of the data presented. Perhaps that is what was done, but it does not come through that way.*

We apologize for the unclear description of the quantifications. We have described the quantification methods in detail in the Materials and methods section(subsection “Image analyses”). With respect to the quantification method and standards, we mainly refer to some articles: “Shim J, Mukherjee T, Banerjee U. 2012. Direct sensing of systemic and nutritional signals by hematopoietic progenitors in *Drosophila*. Nat Cell Biol14: 394-400.”, “Benmimoun B, Polesello C, Waltzer L, Haenlin M. 2012. Dual role for Insulin/TOR signaling in the control of hematopoietic progenitor maintenance in *Drosophila*. Development139: 1713–1717.”, “Gueguen G, Kalamarz ME, Ramroop J, Uribe J, Govind S. 2013. Polydnaviral ankyrin proteins aid parasitic wasp survival by coordinate and selective inhibition of hematopoietic and immune NF-kappa B signaling in insect hosts. PLoS Pathog9:e1003580.” and “Benmimoun B, Polesello C, Haenlin M, Waltzer L. 2015. The EBF transcription factor Collier directly promotes *Drosophila* blood cell progenitor maintenance independently of the niche. Proc Natl Acad Sci U S A112: 9052–9057.”.

For the quantifications of the numbers of crystal cells and PSC cells and ROS indexes, at least 30 lymph glands were scored per genotype. For the quantifications of the areas of P1^+^ cells and the Shg^+^ and domeMESO^+^ cell indexes, at least 30 lymph glands were scored per genotype. For the quantifications of the Col, Dlp, dMyc and Jumu fluorescence signal intensities in the MZ or PSC, at least 20 lymph glands were analyzed. We provide a detailed description about the collection and analysis of images in the Materials and methods section(subsection “Image analyses”).

In this study, the Antp, Hnt and ROS signals were converted to the number of cells, because these three markers are all located in nuclei and show a clearer signal. Moreover, the signal intensity in each cell was similar, and these signal points are relatively scattered and can thus be counted easily. The image acquisition settings were adjusted to avoid over-exposure, but allow observation of a clear scattered signal.

All quantification graphs are presented as box-and-whiskers with the exceptions of Figures 1EE and 9B, which show the quantification of mRNA levels.

*5) In Figure 2., the data look very nice, but a developmental series will be useful in knowing whether the cells are migrating or arising in the wrong site. Overexpression of Antp causes an expansion of the PSC, the reason has never been clear. This again brings up the question of whether ANTP controls JUMU.*

We have added a description of the morphology of the PSC at different larval stages and the colocalization of Antp^+^ PSC cells and *dome-GFP^+^*cells or *Hml-GFP^+^*cells, as shown in Figure 3(subsection “Jumu negatively regulates the proliferation and ectopic expansion of Col^+^ PSC cells”). We observed that the ectopically expanded Antp^+^ PSC cells did not co-localize with any dome-GFP^+^ or Hml-GFP^+^ cells in the MZ and CZ domains. In particular, the ectopic PSC cells adjacent to MZ cells or CZ cells in the early larval stage showed no intermediate cells co-expressing both Antp and GFP. The results indicate that the ectopic PSC cells in the *jumu* mutants are derived from the PSC domain. In addition, previous studies have shown that Antp is only located in the PSC of the lymph gland (Mandal L, Martinez-Agosto JA, Evans CJ, Hartenstein V, Banerjee U. 2007. A Hedgehog- and Antennapedia-dependent niche maintains *Drosophila* hematopoietic precursors. Nature 446: 320–324.), and we also did not find the expression of Antp in MZ using anti-Antp antibody; however, ectopic expansion of the PSC is caused by the loss of *jumu* in the MZ, suggesting that Antp cannot directly control *jumu* in the regulation of PSC morphology.

6) Figure 3. It would be good to see the genotype in panel B stained for MZ and CZ markers.

We have added the results of P1, Hnt, Shg and ROS staining of the *jumu^GE27806^/ jumu^Df2.12^*mutant to Figure 1.

*7) Figure 8., how does panel 1 match with the results in the authors' previous paper?*

Thank you for the advice. dMyc expression is up-regulated in circulating hemocytes of *Hml>UAS-jumu*. Our previous studies showed increased circulating hemocytes and lamellocyte differentiation in *Hml>UAS-jumu*. To further analyze whether these phenomena are related to dMyc, we overexpressed *dmyc* under the control of *Hml-Gal4. Hml>UAS-dmyc* did not present lamellocytes in the lymph glands (the data was shown in Figure 12). This result suggests that the overexpression of *dmyc* in CZ do not affect lamellocyte differentiation in the lymph gland. However, we found that similar to *Hml>UAS-jumu, Hml>UAS-dmyc* also showed increased circulating hemocytes (the data was shown in Figure 12), suggesting dMyc regulates the proliferation of circulating hemocytes. Although dMyc might be one of the factors responsible for the increase in circulating hemocytes in *Hml>UAS-jumu*, some other signaling pathways and elements might also be involved in the proliferation of circulating hemocytes. The role of Jumu in hemocyte proliferation remains to be further investigated.

Author response image 2.**DOI:**
http://dx.doi.org/10.7554/eLife.25094.023

*8) Also Figure 8, the binding assay needs a critical control with mutated binding sites.*

We mutated two positions in each of the three putative FKH-binding sites, the mutated promoter was cloned to the pGL3 vector, and a luciferase reporter gene assay was used to detect their activation. The results suggest that Jumu can endogenously be recruited to the -27/-17 site of the *dmyc* promoter and further strengthen the hypothesis that Jumu positively up-regulates *dmyc* expression by directly binding to the *dmyc* promoter (Figure 8). The main results are described in the third paragraph of the subsection “Jumu cell-autonomously controls the size of the PSC by regulating the expression of dMyc”, and the methods used in the transfection and luciferase reporter gene assay are detailed in subsections “Construction of dmyc promoter reporter plasmid and site-directed mutagenesis” and “Transfection and luciferase reporter gene assay”.

*Reviewer #2:*

*[…] 1) One of the main findings of Jin and Hao is that one-copy loss of Jumu induces expansion of the PSC cells in the lymph gland. However, the phenotype shown by Jumu heterozygote (Figure 2' or H) includes both proliferation and clustering defects of the PSC. Previous study has shown that these two phenomena are regulated by two independent pathways in the PSC; proliferation is regulated by Dpp/dMyc and clustering, by DE-cad/Cdc42 (Morin-Poulard et al., 2015). Authors have demonstrated that expression of Jumu in the MZ rescues both proliferation and clustering phenotypes of the PSC. To support the authors' claim, it will be important to identify genetic correlations in between Jumu in the MZ and two other pathways in the PSC in detail.*

We have analyzed the correlations between Jumu and DE-cad/Cdc42 or Dpp/dMyc in the MZ in the regulation of PSC morphology. The expression levels of Shg (DE-cad) were not reduced in the PSC of *jumu^Df2.12^*compared with the wild-type (Figure 5—figure supplement 1’’). We then performed a rescue experiment to identify the correlation between Jumu and Cdc42. The overexpression of the dominant negative form of Cdc42 (*cdc42-DN*) in the PSC of *jumu ^Df2.12^*did not rescue the increased number and clustering defect of the PSC cell (Figure 5—figure supplement 1). These results suggest that Jumu protein in the MZ controls the clustering morphology of the PSC in a DE-cad/Cdc42-independent manner. In addition, we also found that the expression of Dlp and P-Mad was not reduced in the PSC of *jumu^Df2.12^*compared with the wild-type, suggesting that Jumu does not regulate Dpp signaling in the PSC (Figure 5—figure supplement 1’’, subsection “The loss of jumu in the MZ non-cell-autonomously induces increases in dMyc in the PSC”).

*2) Along the same lines; if Jumu is widely expressed in other tissues including the dorsal vessel, it is possible that Jumu regulates Slit/Robo that in turn regulates both proliferation and clustering of the PSC. To exclude the involvement of Slit/Robo pathway and its indirect regulation, experiments such as Hand-gal4 driving Jumu RNAi or overexpression of Jumu will simply solve this concern.*

We knocked down or overexpressed *jumu* using the *Hand-Gal4* driver to detect PSC cells but did not observe defects in either proliferation or clustering in the PSC (Figure 6—figure supplement 1). Moreover, using *Hand-Gal4* driver to overexpress *jumu* in the *jumu^Df2.12^*mutant did not rescue the PSC phenotype (Figure 4; subsection “The loss of jumu in the MZ non-cell-autonomously induces increases in dMyc in the PSC”).

*3) One-copy loss of Jumu induces expansion of PSC cells in the lymph gland (Figure 2), and this phenotype is recapitulated by Dome-gal4 driving Jumu RNAi (Figure 4). However, downregulation of Jumu in the PSC further reduces the number of PSC cells, while overexpression of Jumu leads to the proliferation of PSC cells. These data indicate that Jumu regulates dMyc in opposite ways in two zones; Jumu in the MZ non-autonomously inhibits dMyc whereas Jumu in the PSC autonomously activates dMyc. However, in the development of the PSC, PSC cells require proliferation at earlier stages while Jumu needs to inhibit the proliferation in later instars. Developmental relevance of Jumu would be also important for understanding the gene function. Supplementing strong evidence that there are changes in the expression levels of Jumu/dMyc throughout larval development, or different phenotypes led by early vs late knockdown of Jumu in different zones would strengthen the developmental function of Jumu/dMyc in the PSC.*

First, we detected the expression pattern of Jumu and dMyc in the lymph gland during larval development. We observed that Jumu and dMyc are detected throughout larval developmental stage and that is expression pattern is similar to that detected in the third-instar lymph gland (Figure 4 and Figure 7—figure supplement 2; subsection “Jumu cell-autonomously controls the size of the PSC by regulating the expression of dMyc”, last paragraph). This result suggests that Jumu protein in the PSC and MZ might consistently balance the dMyc expression levels.

We then used a temporal and regional gene expression targeting system to knock down *jumu* in the PSC or MZ at different developmental stages and detect the PSC numbers. The results show that the PSC cells were reduced or expanded to different degrees at each developmental stage from first to third instar (Figure 7—figure supplement 3, subsection “Jumu cell-autonomously controls the size of the PSC by regulating the expression of dMyc”, last paragraph). These results suggest that Jumu expression in both the MZ and the PSC ensures maintenance of normal PSC development.

Moreover, we hypothesize that Jumu protein located in the MZ mainly prevents the expansion and scattering of PSC cells toward the MZ and indirectly maintains normal MZ development during lymph gland development but is not sufficient to initiate the inhibition of PSC cell proliferation. We also found that the loss of *jumu* in the MZ causes an expansion of the PSC and an increase in dMyc expression and that the overexpression of *jumu* in the MZ did not spontaneously lead to the reduction in the PSC cell number and dMyc expression (Figure 5’ and 5S). In other words, the main function of Jumu in the MZ may be maintenance of the normal development zone margin between the MZ and the PSC as well as maintenance of the homeostasis of each zone of the lymph gland. A detailed discussion is provided in the third paragraph of the Discussion section.

*Reviewer #3:*

*In this study, Hao and Lin explore the role of Jumu in controlling blood progenitor function in Drosophila, specifically in the larval lymph gland. Their findings demonstrate that Jumu maintains prohemocytes by cell-autonomously preventing the overexpression of Col in the MZ and by non-cell-autonomously inhibiting dMyc in the PSC to regulate its size. Their study also studies reveals a requirement for Jumu in cell-autonomously controlling PSC cell proliferation via dMyc. Their findings provide novel insight into the molecular regulation of progenitor maintenance, in particular a novel mechanism in which the progenitor population provides feedback to the cells of the niche/PSC for their own maintenance. The quality of the data is excellent and consistent with established standards for this field. However, there are a few concerns that need to be addressed:*

Expression of Hedgehog in the PSC should be assessed upon overexpression of Jumu in the MZ to determine if the increased differentiation observed is due to loss of this maintenance signal.

We overexpressed or knocked down *jumu* in the MZ and detected the Hh-GFP expression levels. The results showed that the Hh-GFP signal was not changed compared with the control (Figure 6—figure supplement 2; subsection “Jumu cell-autonomously controls the maintenance of blood cell progenitors by regulating the low levels of Col in the MZ”, first paragraph). This result suggests the prohemocyte overdifferentiation due to the overexpression of *jumu* is not related to Hh signaling.

*It cannot be assumed that lamellocytes present in the lymph gland of trans-heterozygote Jumu mutants is due to loss of collier (Discussion, second paragraph). Staining of lymph glands for lamellocytes in which Jumu is removed in specific zones would allow insight into the process.*

The previous description was “Previous studies have reported that the overexpression of collier in the entire lymph gland causes lamellocyte differentiation and reduces the number of crystal cells (Crozatier M, Ubeda JM, Vincent A, Meister M. 2004. Cellular immune response to parasitization in *Drosophila* requires the EBF orthologue collier. PLoS Biol2: E196). Thus, Jumu might control the differentiation of lamellocytes by negatively regulating *col* expression in the entire lymph gland”.

According to the suggestion, we have knocked down *jumu* or overexpressed *col* in different zones of the lymph gland (CZ, MZ and PSC) and did not observe obvious increases in the lamellocyte numbers in the lymph gland; in fact, less than 20% and 10% of the lymph glands showed a few lamellocytes ((Figure 9—figure supplement 1). However, *jumu* knockdown and *col* overexpression in the entire lymph gland caused obvious increases in lamellocytes and reductions in crystal cells. Moreover, the knockdown of *col* in *Srp>jumu RNAi* can inhibit lamellocyte differentiation (Figure 9; subsection “The loss of jumu in the entire lymph gland leads to the generation of lamellocytes through the activation of Toll signaling”, last paragraph). These results indicate that the loss of *jumu* in the entire lymph gland might induce lamellocyte differentiation by up-regulating the expression of Col.

*Dome-gal4 is expressed outside of the MZ in many other organs and the authors cannot rule an effect external to the signal responsible for the observed effect of inhibiting PSC expansion in Jumu mutants. This should be stated.*

With the exception of the cardiac tube, the other organs involved in PSC morphology have not been reported. However, we knocked down or overexpressed *jumu* using the *Hand-Gal4* driver (cardiac tube) to detect the PSC cells and did not neither proliferation nor clustering in the PSC (Figure 6—figure supplement 1 A-1C).

To further confirm our conclusion, we used a pan lymph gland-specific *Srp-Gal4* or MZ-driver *TeplV-Gal4* to over express *jumu* in the *jumu* mutant and found that this overexpression can rescue the PSC defect (Figure 4 and Figure 5). However, the overexpression of *jumu* using Antp-Gal4 (PSC driver) and Hml-Gal4 (CZ driver) did not rescue the PSC defect in the *jumu* mutant (Figure 4). Therefore, the results indicate that Jumu protein in the MZ inhibits PSC expansion in the lymph gland.

*It would be useful to have clones of Jumu made in which the cell autonomous effects observed can be established, particularly given that the levels of Jumu may have different effects on cell maintenance and differentiation.*

Thank you for this suggestion. We have added the results of a clonal analysis using HHLT-Gal4 to Figure 6—figure supplement 3in the second paragraph of the subsection “Jumu cell-autonomously controls the maintenance of blood cell progenitors by regulating the low levels of Col in the MZ” and in the first paragraph of the subsection “Jumu cell-autonomously controls the size of the PSC by regulating the expression of dMyc”.

We previously indicated two cell-autonomous functions of Jumu in lymph gland development: First, Jumu protein in the MZ can cell-autonomously promote prohemocyte differentiation (Figure 6), and second, Jumu protein in the PSC can cell-autonomously promote PSC cell proliferation (Figure 7). The clonal analysis confirmed these conclusions. First, clonal cells in which *jumu* was knocked down rarely overlapped with P1^+^ cells (Figure 6—figure supplement 3*’’*), but most P1^+^ cells overlapped with clonal cells overexpressing *jumu* (Figure 6—figure supplement 3’’), suggesting that Jumu autonomously promotes hemocyte differentiation. Second, the clonal knock down of *jumu* in PSC cells can reduce the number of PSC cells (Figure 6—figure supplement 3); however, the clonal cells overexpressing *jumu* in the PSC tended to increase the PSC cell number (Figure 6—figure supplement 3). Altogether, these results further confirm that Jumu cell-autonomously promotes PSC cell proliferation.

*In addition, the following need to be corrected prior to publication:*

*Figure 1: the ROS staining looks poorly done.*

Fluorescence staining for ROS levels is typically performed using DHE (dihydroethidium), which is a redox-sensitive dye that intercalates into cellular DNA when oxidized. Thus, the fluorescence signal does not look as clear as that obtained through antibody staining. Mondal et al.provided a description of a DHE staining protocol for the lymph gland, which is similar to that used in the present study.

Figure 2: these experiments need labeling with a MZ marker like shg or ptc.

We have added Shg staining to Figure 2and describe these results in the subsection “Jumu negatively regulates the proliferation and ectopic expansion of Col^+^ PSC cells”.

*Figure 2: it is not clear if the ectopic Antp cells are derived from the PSC or the result of a homeotic transformation in which an ectopic PSC is generated. The fact that Dome>UAS-Jumu rescues the phenotype of ectopic PSC may have to do with Jumu preventing the PSC fate in the MZ (domeless is expressed very early in lymph gland development) rather than the PSC increasing in size.*

We thank the reviewer for these constructive comments. We analyzed PSC cells at different larval stages but did not observe the colocalization of Antp^+^ PSC cells with *dome-GFP^+^*cells or *Hml-GFP^+^*cells (Figure 3, subsection “Jumu negatively regulates the proliferation and ectopic expansion of Col^+^ PSC cells”, last paragraph). These results indicate that the ectopic PSC cells in the *jumu* mutants are derived from the PSC domain.

*Figure 2: The quantification panel can go in supplementary to maintain symmetry for the figure.*

The construction and order of images in this figure have been adjusted (Figure 2).

*Figure 3: the Jumu staining looks cytoplasmic while clearly nuclear with some cytoplasmic in the PSC. Have the authors looked at the developmental pattern of expression of Jumu to determine if it is ever nuclear in the lymph gland outside of the PSC? The developmental pattern of Jumu expression is essential to understand its function.*

We have detected the expression pattern of Jumu in the lymph gland throughout larval development and found that Jumu is always expressed in the entire lymph gland, with markedly higher expression in the PSC, and that it is mainly localized in the nuclei (Figure 4’). Previous studies have shown that Jumu is located in the nuclei of syncytial embryos, is present in all nuclei of the cellular blastoderm, and is predominantly expressed in the brain lobes, in the segmented CNS and in elements of the PNS. In third-instar larvae, the most prominent expression of Jumu is in imaginal discs, the CNS, the ring gland and the hindgut (Cheah, P.Y., Chia, W., Yang, X., 2000. Jumeau, a novel *Drosophila* winged-helix family protein, is required for generating asymmetric sibling neuronal cell fates. Development. 127, 3325-3335; Sugimura, I., Adachi-Yamada, T., Nishi, Y., Nishida, Y., 2000. A *Drosophila* Winged-helix nude (Whn)-like transcription factor with essential functions throughout development. Dev Growth Differ.42, 237-248). We successfully prepared anti-jumu polyclonal antibodies, and the results were published in the Immunological Journal of China (2011, 27:1038-1042). The specificity and titer of the antibodies were detected by Western blotting and immunostaining. The subcellular localization and tissue-specific expression of Jumu protein were analyzed by immunostaining. The results showed endogenous Jumu protein in the nuclei of blood cells and the fat body, and this signal was not be detected in *jumu* mutants flies. In this study, we show that Jumu is localized in the nuclei of lymph gland cells and that the subcellular localization of Jumu does not change during larval development.

Figure 3: images too blurry.

The images in Figure 3have been replaced’

Figure 3: can go in supplementary.

This figure has been renumbered as Figure 4.

Figure 4: the figure is missing dome-gal4, UAS-Jumu, UAS-col with Antp immunostaining.

Both *dome>UAS-jumu* and *dome>UAS-col* did not cause the PSC defect; thus, we do not think that it is necessary to overexpress *col* in *dome>UAS-jumu* to rescue the PSC cell numbers.

*Figure 9: the schematic model is misleading since the trans heterozygote must include a hypomorphic mutation and not a complete loss of function to allow survival into adulthood. Please modify the description to trans heterozygote.*

We apologize for the incorrect description. We used double heterozygotes instead of trans-heterozygotes.